# GnRH pulse generator activity in mouse models of polycystic ovary syndrome

**Ziyue Zhou[1], Su Young Han[1], Maria Pardo-Navarro[1], Ellen G Wall[1], Reena Desai[2], Szilvia Vas[1], David J Handelsman[2], Allan E Herbison[1]\***

[1]Department of Physiology, Development and Neuroscience, University of Cambridge, Cambridge, United Kingdom; [2]ANZAC Research Institute, University of Sydney, Sydney, Australia

## eLife Assessment

This **important** study reports findings on the GnRH pulse generator's role in androgen-exposed mouse models, providing further insights into PCOS pathophysiology and advancing the field of reproductive endocrinology. The experimental data were collected using cutting-edge methodologies and are **solid**. The findings, while interesting, are primarily applicable to mouse models, and their translation to human physiology requires cautious interpretation and further validation. This work will be of interest to endocrinologists and reproductive biologists.

**\*For correspondence:**
aeh36@cam.ac.uk

**Competing interest:** The authors declare that no competing interests exist.

**Abstract** One in ten women in their reproductive age suffer from polycystic ovary syndrome (PCOS) that, alongside subfertility and hyperandrogenism, typically presents with increased luteinizing hormone (LH) pulsatility. As such, it is suspected that the arcuate kisspeptin (ARN$^{KISS}$) neurons that represent the GnRH pulse generator are dysfunctional in PCOS. We used here in vivo GCaMP fiber photometry and other approaches to examine the behavior of the GnRH pulse generator in two mouse models of PCOS. We began with the peripubertal androgen (PPA) mouse model of PCOS but found that it had a reduction in the frequency of ARN$^{KISS}$ neuron synchronization events (SEs) that drive LH pulses. Examining the prenatal androgen (PNA) model of PCOS, we observed highly variable patterns of pulse generator activity with no significant differences detected in ARN$^{KISS}$ neuron SEs, pulsatile LH secretion, or serum testosterone, estradiol, and progesterone concentrations. However, a machine learning approach identified that the ARN$^{KISS}$ neurons of acyclic PNA mice continued to exhibit cyclical patterns of activity similar to that of normal mice. The frequency of ARN$^{KISS}$ neuron SEs was significantly increased in algorithm-identified 'diestrous stage' PNA mice compared to controls. In addition, ARN$^{KISS}$ neurons exhibited reduced feedback suppression to progesterone in PNA mice and their gonadotrophs were also less sensitive to GnRH. These observations demonstrate the importance of understanding GnRH pulse generator activity in mouse models of PCOS. The existence of cyclical GnRH pulse generator activity in the acyclic PNA mouse indicates the presence of a complex phenotype with deficits at multiple levels of the hypothalamo-pituitary-gonadal axis.

## Introduction

Polycystic ovary syndrome (PCOS) is one of the most common causes of female infertility with variable reproductive and metabolic phenotypes (*Bozdag et al., 2016*; *Lizneva et al., 2016*). Many women with PCOS have an elevated body mass index but up to 30% of patients do not, resulting in PCOS being further classified into obese and lean subtypes (*Nestler, 1997*; *Kumari et al., 2005*; *Toosy et al., 2018*). Although androgen excess, oligo-menorrhea/anovulation, and the existence of

polycystic-like ovarian morphology are the key diagnostic indicators for PCOS, a common secondary feature is increased luteinizing hormone (LH) pulsatility (*Waldstreicher et al., 1988*, *Rotterdam ESHRE/ASRM-Sponsored PCOS consensus workshop group, 2004*). Basal and pulsatile LH secretion can be up to threefold higher in both lean and obese PCOS women and are considered to be a significant contributor to their subfertility (*Morales et al., 1996*, *Van Dam et al., 2002*; *Phylactou et al., 2021*). A presumed increase in the frequency of pulsatile gonadotropin-releasing hormone (GnRH) secretion in PCOS women results in an increased LH/follicle-stimulating hormone (FSH) ratio (*Wildt et al., 1981*; *Dalkin et al., 1989*). This is thought to drive excess ovarian androgen production that, in turn, impairs estrogen and progesterone feedback regulation of GnRH secretion (*Pastor et al., 1998*; *Chhabra et al., 2005*; *Moore et al., 2013*; *Moore et al., 2015*).

It has become clear that the arcuate nucleus kisspeptin (ARN$^{KISS}$) neurons represent the mammalian GnRH pulse generator (*Clarkson et al., 2017*; *Herbison, 2018*; *Goodman et al., 2022*). These neurons exhibit brief episodes of synchronized activity that release kisspeptin on the distal projections of GnRH neurons to drive pulsatile GnRH secretion into the portal vasculature (*Herbison, 2021*; *Liu et al., 2021*). Recent studies indicate that the synchronized activity amongst ARN$^{KISS}$ neurons required for pulsatile hormone secretion is generated by local glutamatergic transmission that is subsequently modulated by neurokinin B and dynorphin (*Qiu et al., 2016*; *Han et al., 2023*; *Morris and Herbison, 2023*). While the ARN$^{KISS}$ neurons are the primary site of estrogen negative feedback suppressing pulsatile GnRH secretion (*McQuillan et al., 2022*), the mechanism of progesterone feedback remains unclear but could also involve the ARN$^{KISS}$ neurons (*Goodman et al., 2011*). As such, it is thought that dysfunction within the ARN$^{KISS}$ neuron pulse generator may be responsible for the elevated LH pulse frequency observed in PCOS (*Walters et al., 2018*; *Coutinho and Kauffman, 2019*; *Rodriguez Paris and Bertoldo, 2019*; *Ruddenklau and Campbell, 2019*).

We have developed GCaMP fiber photometry methodologies that allow the activity of the ARN$^{KISS}$ neuron pulse generator to be monitored at high temporal resolution in freely behaving mice (*Han et al., 2019*; *McQuillan et al., 2019*; *Vas et al., 2024*). Using this and other approaches, we report here a detailed investigation into pulse generator activity in two widely used mouse models of PCOS including the peripubertal androgen (PPA) model of 'obese PCOS', and prenatal androgen (PNA) model of 'lean PCOS' (*Stener-Victorin et al., 2020*).

## Results

### Peripubertal androgen (PPA) mice

### Peripubertal dihydrotestosterone (DHT)-treated animals exhibit significantly increased body weight with acyclic estrous cycles

Female 129S6Sv/Ev C57BL/6 *Kiss1$^{Cre/+}$*,GCaMP6s mice exposed to DHT from 3 weeks of age (PPA: n=4) started to show a significant increase in weight 4 weeks after capsule implantation compared to controls with empty capsules (control: n=5; Sidak's multiple comparisons test; *Figure 1—figure supplement 1A*). The PPA animals were completely acyclic with vaginal cytology only ever showing the diestrous stage (*Figure 1—figure supplement 1B-D*).

### Peripubertal exposure to DHT results in a slowed GnRH pulse generator

Abrupt increases in arcuate nucleus kisspeptin (ARN$^{KISS}$) neuron population activity, termed synchronization events (SEs), are observed with GCaMP fiber photometry in mice (*Clarkson et al., 2017*; *Han et al., 2019*; *McQuillan et al., 2019*). Both adult PPA and control mice recorded in diestrus for 24 hr exhibited ARN$^{KISS}$ neuron SEs (*Figure 1A and B*). The frequency of SEs in PPA mice (n=4) was approximately half that of controls (n=5) with SE frequency/hour showing an inhibitory trend (p=0.0714, Mann-Whitney U test; *Figure 1C*) and SE interval significantly increased (p=0.0159, Mann-Whitney U test; *Figure 1D*). The SE interval frequency distribution revealed a shift to the right for PPA mice with a clustering within the 70–90 min bin centers (*Figure 1E*). To compare the dynamics of SEs from control and PPA animals, continuous (10 Hz) photometry recordings were performed in diestrus to obtain high temporal resolution profiles (*Figure 1F*). These profiles were the same between control (n=5) and PPA (n=4) animals with SE duration, and both upswing and downswing durations, calculated from the full-width half maximum (FWHM) points, being not significantly different (*Figure 1G–H*).

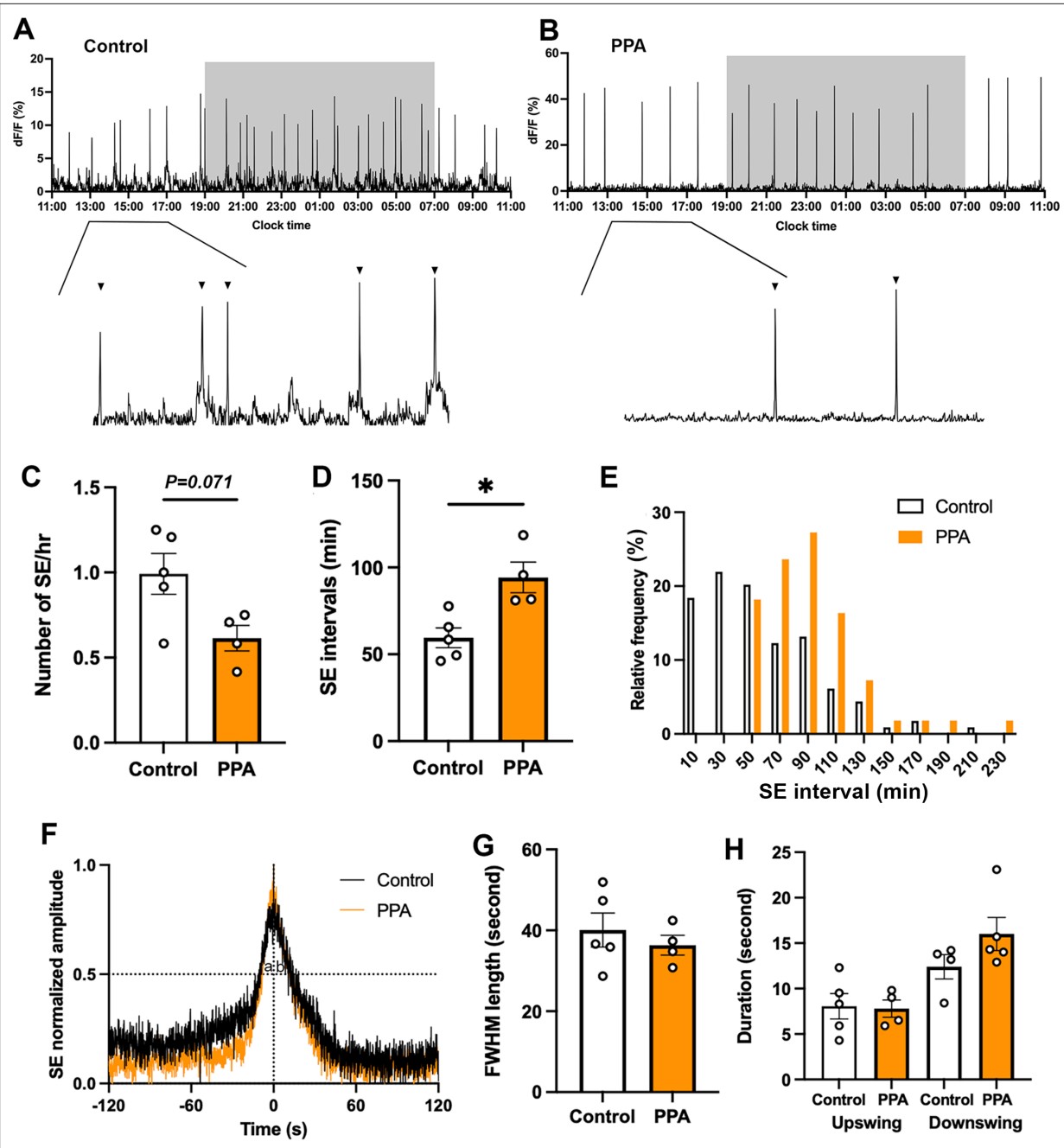

**Figure 1.** Slowerpulse generator activity in PPA animals. Representative 24 hr photometry recordings showing synchronization events (SEs) observed in diestrous (**A**) control and (**B**) PPA females with the light-off period (19:00-07:00) represented by the shaded area and expanded views of the traces (13:00-17:00) given below. Triangles point to identified SEs. (**C**) SE frequency and (**D**) SE intervals in control (n=5) and PPA (n=4) mice. Mann-Whitney U tests. (**E**) Frequency histograms showing relative SE frequencies occurring in 20 min bins, calculated separately for controls (white, n=114, 5 mice) and PPA (orange, n=55, 4 mice). X-axis represents the bin centers. (**F**) Continuous recordings at 10 Hz sampling rate showing normalized profile of SE overlaid from control (black, 7 SEs from 5 animals) and PPA (orange, 5 SEs from 4 animals). (**G**) FWHM length of control (n=5) and PPA (n=4) animals. Mann-Whitney U test. (**H**) Durations of upswing and downswing for control (n=5) and PPA (n=4) animals, respectively. Kruskal-Wallis test followed by Dunn's multiple comparisons test. Data show mean ± SEM. Each circle is an individual animal. * p<0.05.

The online version of this article includes the following figure supplement(s) for figure 1:

**Figure supplement 1.** Peripubertal androgen (PPA) treatment causes an increase in body weight and a loss ofestrous cyclicity.

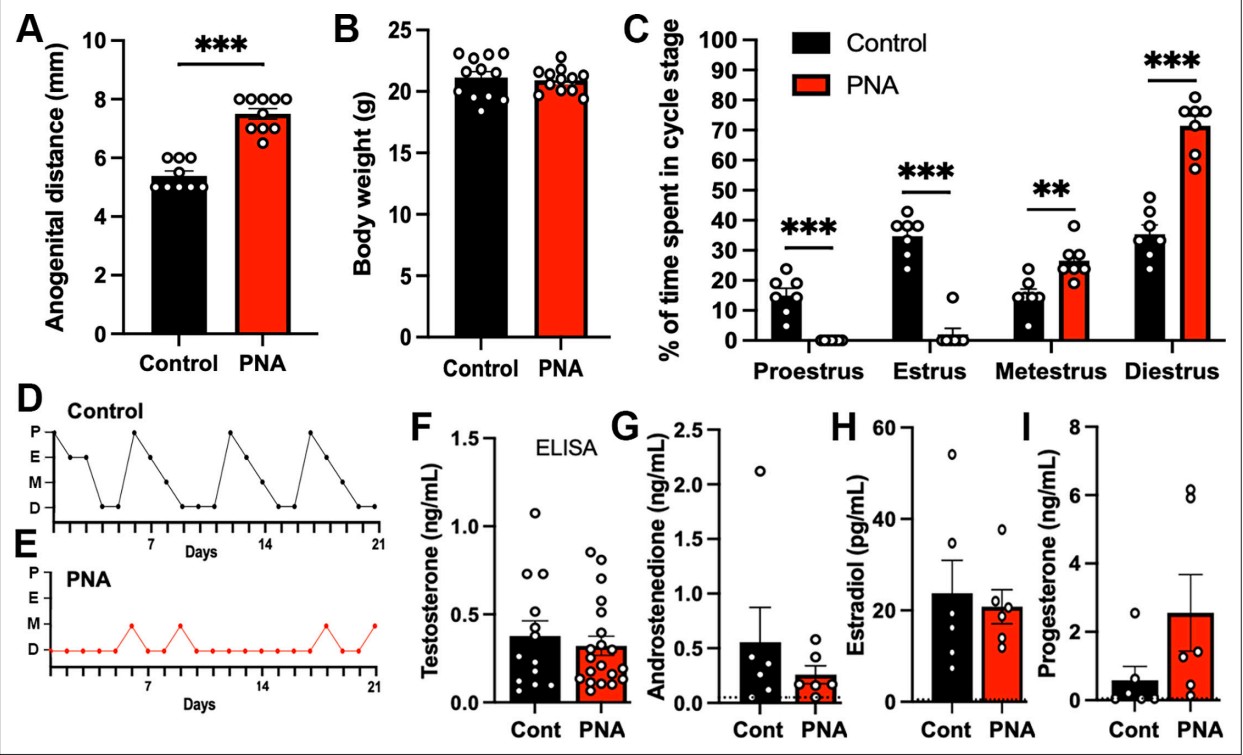

**Figure 2.** Prenatal androgen exposure leads to increased anogenital distance and disrupted estrous cyclicity. (**A**) Anogenital distance (control: n=9; PNA: n=10) and (**B**) Body weight of control and PNA animals (n=12 per group). Mann-Whitney U tests. (**C**) Proportion of time spent at each estrous cycle stage over 21 days (n=7 per group). Two-way ANOVA followed by Sidak's post-hoc tests. (**D–E**) Representative graphs of estrous cycle pattern in (**D**) control and (**E**) PNA females. P, proestrus; E, estrus; M, metestrus; D, diestrus. (**F**) Serum testosterone level of control (n=13) and PNA (n=20) animals in diestrus measured by enzyme-linked immunosorbent assay (ELISA). The limit of detection for testosterone ELISA is 0.066 ng/mL. (**G–I**) Plasma levels of (**G**) androstenedione, (**H**) estradiol, and (**I**) progesterone in control and PNA animals measured using liquid chromatography-mass spectrometry (LC-MS). Blood samples were collected at 10:00 on diestrus (n=6 per group). Dotted lines represent the limit of detection for each hormone. Androstenedione: 0.05 ng/mL; estradiol: 0.50 pg/mL; progesterone: 0.05 ng/mL. All samples were below limit of detection for testosterone LC-MS measurement (<0.01 ng/mL). Mann-Whitney U tests. Data show mean ± SEM. Each circle is an individual animal. ** p<0.01, *** p<0.001.

The online version of this article includes the following figure supplement(s) for figure 2:

**Figure supplement 1.** Slightly increased kisspeptin immunofluorescence in the rostral arcuate nucleus of PNA mice.

## Prenatal androgen (PNA) mice

### The PNA mouse model exhibits highly disordered estrous cyclicity

Female mice originating from dams treated with DHT on gestational days 16, 17, and 18 were found to have a significantly increased anogenital distance (n=10; p<0.001, Mann-Whitney U test) compared with controls (n=9; *Figure 2A*). Body weight was not different between PNA and control animals at 10 weeks of age (n=12 per group; p=0.6602, Mann-Whitney U test; *Figure 2B*). The PNA mice exhibited very disordered estrous cycles remaining mostly in persistent diestrus (71 ± 3%) with occasional excursions to metestrus. The proestrous stage was never encountered and estrus was observed rarely (*Figure 2C–E*).

### Circulating gonadal steroid hormone levels are not different between control and PNA animals

As androgen excess is one of the hallmarks of PCOS, we initially measured serum testosterone levels in control and PNA animals using a commercial testosterone enzyme-linked immunosorbent assay (ELISA; *Risal et al., 2021*). This did not detect any difference in serum testosterone levels between control (n=13) and PNA (n=20) animals in diestrus (p=0.8203, Mann-Whitney U test; *Figure 2F*). Given concerns over the sensitivity and specificity of the steroid hormone ELISAs (*Handelsman et al., 2015*), we progressed to using ultrasensitive liquid chromatography-mass spectrometry (LC-MS; *Handelsman*

*et al., 2020*) to measure the circulating levels of testosterone, androstenedione, estradiol, and progesterone in diestrous PNA and control mice (n=6 per group). No significant differences (Mann-Whitney U tests) were detected for androstenedione, estradiol, or progesterone while all samples were below the limit of detection for testosterone (<0.01 ng/mL) (*Figure 2G–I*).

## Kisspeptin, neurokinin B, and dynorphin immunoreactivity are relatively unchanged within the ARN of PNA mice

Previous studies have found variable effects of PNA treatment on kisspeptin, neurokinin B (NKB), and dynorphin protein and mRNA expression within the ARN (*Barnes et al., 1989*, *Yan et al., 2014*; *Osuka et al., 2016*; *Gibson et al., 2021*; *McCarthy et al., 2022*; *Moore et al., 2021*). To re-examine this issue, we assessed software-identified immunoreactive signal intensity for all three peptides in the ARN of control (n=4–8) and PNA (n=4–6) mice (*Figure 2—figure supplement 1A-C*). This revealed a significant but modest elevation in kisspeptin immunofluorescence within the rostral ARN (p=0.0187, Sidak's multiple comparisons test) in PNA mice but not elsewhere in the ARN (*Figure 2—figure supplement 1A*). No significant changes were detected for NKB or dynorphin immunoreactivity in any region of the ARN (*Figure 2—figure supplement 1B, C*).

## PNA animals exhibit highly variable patterns of ARN$^{KISS}$ neuron SEs

Both control and PNA mice exhibited robust SEs across 24 hr recording periods (*Figure 3A–D*). Control mice recorded in diestrus had approximately 0.8 SEs per hour (*Figure 3E*) with an SE interval of 74.75±5.71 min (*Figure 3F*) and a normal-like distribution of SE intervals (*Figure 3G*). Unexpectedly, diestrous PNA mice exhibited highly variable patterns of ARN$^{KISS}$ neuron activity that could appear similar to diestrous controls (*Figure 3C*) or very different (*Figure 3D*) with episodes of multi-peak SEs (mpSEs). However, overall SE frequency and intervals were not different between PNA (n=6) and control (n=7) animals (*Figure 3E and F*; Mann-Whitney U tests). The frequency distribution of inter-SE intervals in PNA mice exhibited a peak interval cluster around the 10 min bin center (39%) with a gradual tail off in SE intervals (*Figure 3G*). Continuous (10 Hz) photometry recordings in diestrus did not reveal any difference in the profiles of single SEs between controls (n=7) and PNA (n=6) animals (*Figure 3H–J*).

## PNA mice have elevated mean LH secretion but normal LH pulse frequency

We have previously found a near perfect relationship between ARN$^{KISS}$ neuron SEs and pulsatile LH secretion in male and female mice (*Han et al., 2019*; *McQuillan et al., 2019*). To assess this relationship in PNA mice, ARN$^{KISS}$ neuron recordings were made with tail-tip blood samples taken at 12 min intervals until an SE was detected after which the blood sampling rate was switched to 6 min intervals. We observed a perfect correlation between SEs and pulsatile LH secretions in both control and PNA mice: every SE was followed by an LH pulse, and no LH pulses were detected without a preceding SE (*Figure 4A and C*). Temporal analysis of SEs and LH release indicated that the SE peak precedes each LH peak by 7.7±0.6 min and 6.6±0.4 min in control (n=7 from 4 mice) and PNA (n=7 from 5 mice) animals (p=0.190, Mann-Whitney U test), respectively (*Figure 4B and D*).

The absence of significant differences in ARN$^{KISS}$ neuron SE frequency in PNA mice and their tight correlation with pulsatile LH secretion indicated that pulsatile LH secretion may not be different between PNA and control mice. Prior studies have found conflicting data with one finding no difference in LH pulse frequency (*McCarthy et al., 2022*) and another reporting increased LH pulsatility (*Moore et al., 2015*). To re-address this we undertook 4 hr tail-tip blood sampling at 10 min intervals to assess pulsatile LH secretion in wild-type C57BL6/J diestrous-stage PNA (n=8) and control (n=7) mice. Mean LH levels were found to be significantly elevated in PNA mice (p=0.0205, Mann-Whitney U test; *Figure 4E–G*) but the variable LH pulse frequency (p=0.1843, Mann-Whitney U test) and amplitude (p=0.4630, Mann-Whitney U test) were unchanged compared to controls (*Figure 4H, I*).

To evaluate whether the response of pituitary gonadotrophs to GnRH was altered, PNA and control mice were given GnRH (200 ng/kg, i.p.) at the end of the 4 hr tail-tip bleeding, and two further tail-tip blood samples collected for LH measurement at 10 min intervals. Both control (n=7) and PNA (n=11) mice showed elevated LH levels in response to exogenous GnRH (*Figure 4J*). There was no significant main effect of PNA treatment (F(1,16)=0.6752, p=0.4233). However, PNA mice showed a small but significantly lower level of LH 20 min after GnRH injection (p=0.0152, Sidak's multiple comparisons

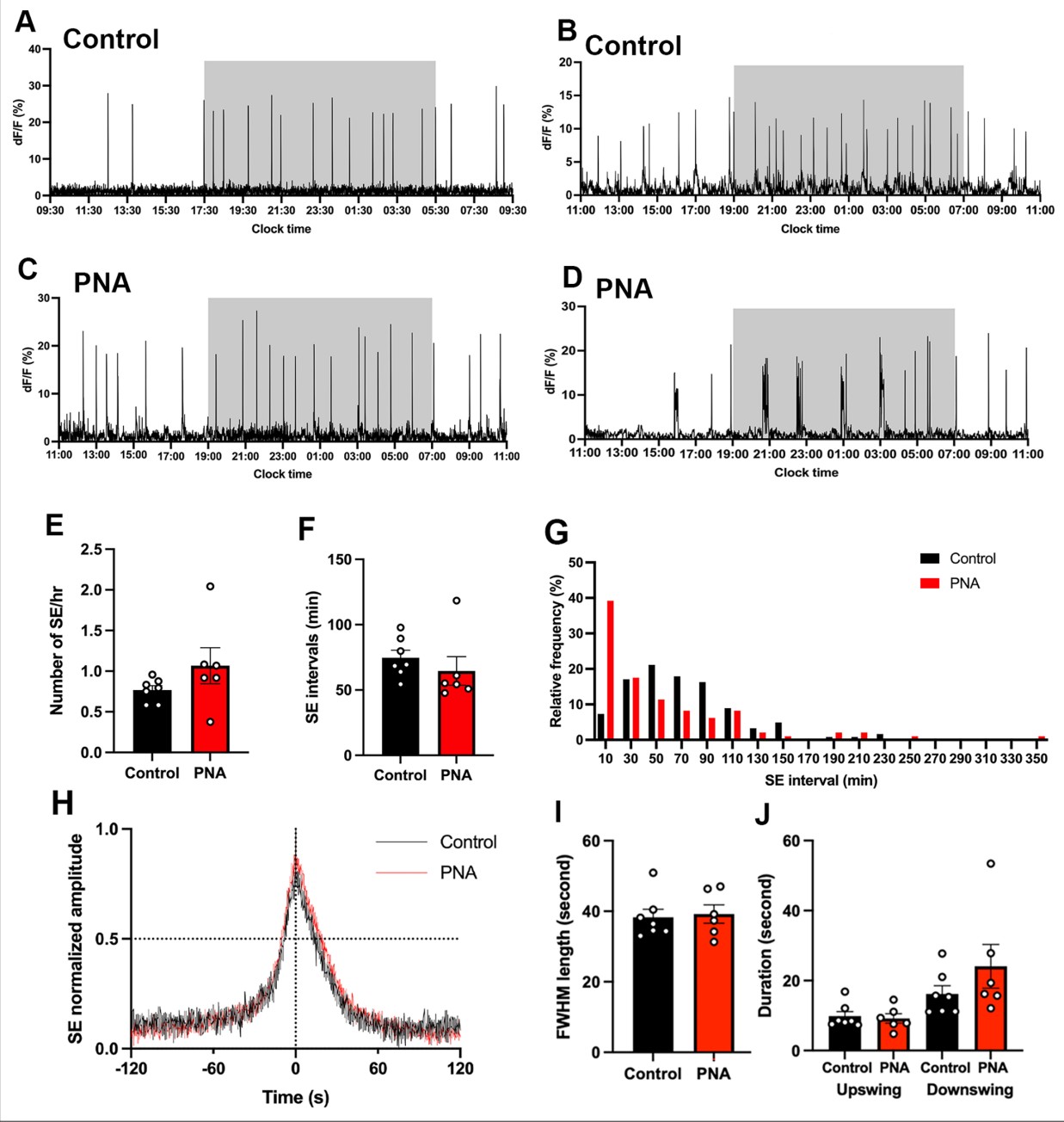

**Figure 3.** PNA animals exhibit highly variable patterns of ARN<sup>KISS</sup> neuron SEs. Representative 24 hr photometry recordings showing SEs observed in (**A,B**) two control and (**C,D**) two PNA females with the light-off period (17:30-05:30 or 19:00-07:00) represented by the shaded area. Recordings were performed when vaginal cytology indicated diestrus. (**E**) SE frequency and (**F**) SE intervals in control (n=7) and PNA (n=6) mice. Mann-Whitney U tests. (**G**) Frequency histograms showing relative SE frequencies occurring in 20 min bins, calculated separately for controls (black, n=123, 7 mice) and PNA (red, n=98, 6 mice). X-axis represents the bin centers. Multi-peak SEs (mpSE) with peaks occurring within 160 s were considered as one SE with interval calculated from the first peak. (**H**) Continuous (10 Hz) recording showing normalized mean profile of SE overlaid from control (black, 22 SEs from 7 animals) and PNA (red, 20 SEs from 6 animals). (**I**) FWHM length (upswing +downswing) in control (n=7) and PNA (n=6) animals. Mann-Whitney U test. (**J**) Durations of FWHM upswing and downswing for control (n=7) and PNA (n=6) animals, respectively. Kruskal-Wallis test followed by Dunn's multiple comparisons test. Data show mean ± SEM. Each circle is an individual animal.

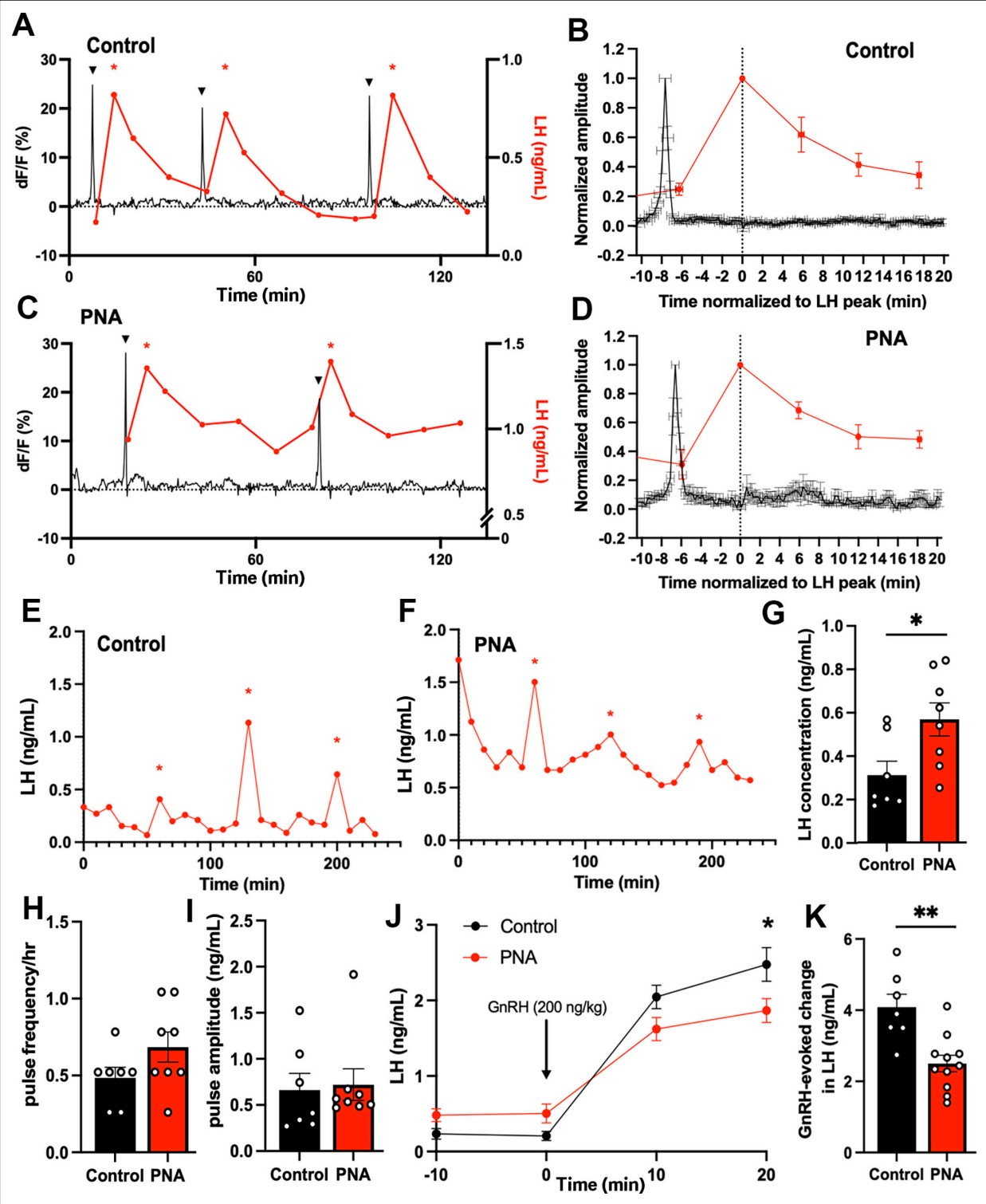

**Figure 4.** Increased total LH concentration but normal LH pulse frequency in PNA animals. Representative examples from (**A**) control and (**C**) PNA mice showing the relationship of SEs (black) with LH secretions (red). Triangles and red asterisks indicate identified SEs and LH pulses, respectively. Normalized LH secretion plotted against normalized SEs in (**B**) control and (**D**) PNA animals. The amplitudes of SEs and serum LH levels were normalized to their peaks, with time 0 being the peak of LH in control (n=7 from 4 mice) and PNA (n=7 from 5 mice) animals. Representative examples of LH pulse profiles in (**E**) control and (**F**) PNA animals. Red asterisks indicate identified LH pulses. (**G**) Total LH concentration, (**H**) LH pulse frequency per hour, and (**I**) LH pulse amplitude in diestrous control (n=7, with 13 LH pulses) and PNA (n=8, with 21 LH pulses) mice across 4 hr sampling at 10 min intervals. Mann-Whitney U tests. (**J**) Change in LH levels following an intraperitoneal injection of GnRH (200 ng/kg, i.p.) in diestrous control (n=7) and PNA (n=11)

*Figure 4 continued on next page*

*Figure 4 continued*

mice. An asterisk above the 20 min data point denotes significance between LH levels in control and PNA using two-way repeated-measure ANOVA followed by Sidak's post-hoc tests. (**K**) GnRH-evoked changes in LH levels in control (n=7) and PNA (n=11) mice were calculated by the difference in LH levels before (–10 min and 0 min) and after (10 min and 20 min) GnRH injections. Mann-Whitney U test. Data show mean ± SEM. Each circle is an individual animal. * p<0.05, ** p<0.01.

test). The total GnRH-evoked increment in LH release was significantly lower in PNA mice compared to controls (p=0.0028, Mann-Whitney U test, *Figure 4K*).

## The pulse generator exhibits cyclical activity in PNA mice

The patterns of ARN[KISS] neuron SEs recorded from acyclic diestrous PNA mice were highly variable and could exhibit typical diestrous-like activity but also characteristics such as mpSEs (*Figure 3D*) that are only usually observed during estrus and metestrus (*Vas et al., 2024*). This suggested that, despite the acyclical state of the PNA mice, the pulse generator may still be undergoing cycles of activity. To assess this, we conducted long-duration photometry recordings in both PNA and control mice for 4 consecutive days starting on proestrus for controls and any day for the acyclic PNA mice. We have recently found that k-means clustering, an machine learning-based approach, provides a very sensitive characterization of ARN[KISS] neuron activity across the estrous cycle (*Vas et al., 2024*). This involves analysis of SEs based on five parameters: the frequency of regular single-peak SEs; the standard deviation of inter-SE intervals for regular SEs; and the number, duration, and profiles of mpSEs (*Figure 5A*).

All control animals showed typical cycling patterns of ARN[KISS] neuron activity over the 4-day recording period (n=5, two representative examples shown in *Figure 5C*). The most pronounced features of this cyclicity are the suppression of all SE activity during late proestrus to early estrus (represented by Cluster_0 activity) with a gradual return in SEs during late estrus (Cluster_1). This is often followed by the appearance of mpSEs (Cluster_3/4) during the estrus to metestrus transition. The activity patterns then become the typical diestrous single SEs with variable frequencies (Cluster_1 and _2; *Figure 5C*). Surprisingly, we observed that the ARN[KISS] neurons in PNA mice also exhibited variable shifting SE profiles over the 4-day recordings despite vaginal smears remaining in metestrus or diestrus throughout (n=4, two examples shown in *Figure 5D*). These PNA mice exhibited a cyclical pattern of activity reminiscent of control mice with a period of slow Cluster_0 activity directly followed by an mpSEs period (Cluster_3/4) and then regular diestrous activity (Cluster_1 and _2; *Figure 5D*).

Overall, quantification of the time spent in each cluster pattern over the four days was remarkably similar between control (n=5) and PNA (n=4) mice (*Figure 5B*). The proportion of Cluster_0 (PNA 21%; control 21%) and Cluster_1 (PNA 44%; control 42%) activity was almost identical with the major trend being a nonsignificant increased time (p>0.05, Sidak's multiple comparisons test) spent in mpSE Cluster_3/4 activity (PNA 20%; control 12%) at the expense of the fast regular SE Cluster_2 pattern (PNA 15%; control 25%; *Figure 5B*).

## PNA mice have increased diestrous-like GnRH pulse generator frequency

Given that our earlier recordings from PNA mice (*Figure 3C and D*) were unavoidably taken from random times of the pulse generator cycle, we used the machine learning algorithm above to identify 'diestrous-like' (Cluster_1 and _2) episodes of ARN[KISS] neuron SE activity in PNA mice and compared this with diestrous controls (also defined by the same parameters). In this case, the frequency of SEs was 48% greater in PNA mice (n=6) compared to control animals (n=7; p=0.0029, Mann-Whitney U test; *Figure 6C*) and was also reflected in a significantly lower inter-SE interval (p=0.0047, Mann-Whitney U test; *Figure 6D*). The frequency distribution of inter-SE intervals in PNA mice was slightly shifted to the left compared to control mice so that majority of PNA SE intervals were in 10–70 min bin compared to the 30–90 min bin centers in control mice (*Figure 6E*).

## PNA mice have impaired progesterone negative feedback

Progesterone negative feedback is thought to be abnormal in women with PCOS (*Pastor et al., 1998*; *Chhabra et al., 2005*) and PNA mice (*Moore et al., 2015*). We assessed the response of ARN[KISS] neuron activity to vehicle or progesterone (4 mg/kg, i.p.) in PNA and control mice with

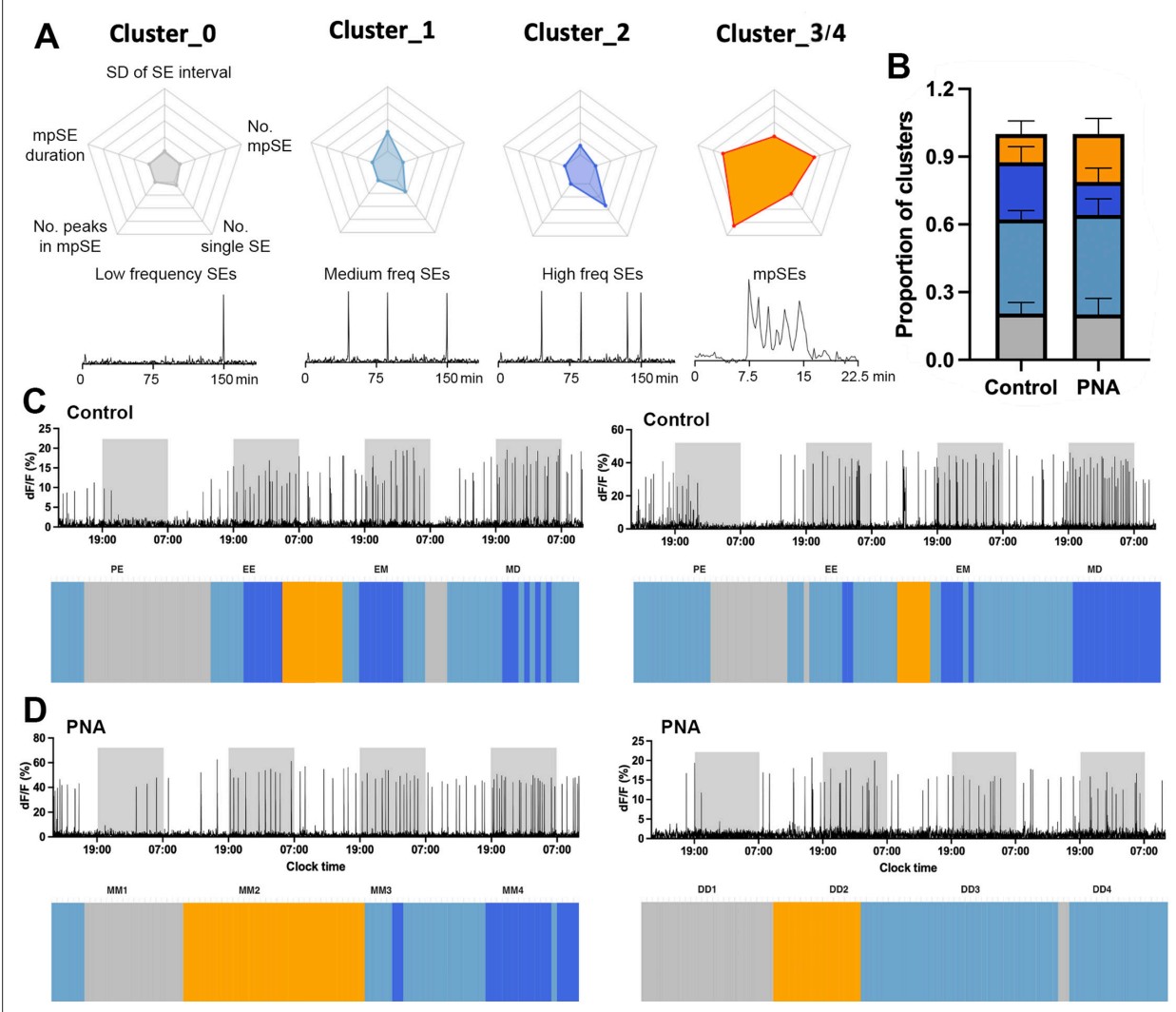

**Figure 5.** PNA animals exhibit cycling pulse generator activity. (**A**). Cluster centroid values for normalized parameters used in k-means clustering with a schematic plot of corresponding photometry recordings patterns. The parameters used for cluster assignments are labelled in Cluster_0 with the following axes: (top) standard deviation (SD) of SE intervals, (top right) number of multi-peak SEs (mpSEs), (bottom right) number of single SEs, (bottom left) number of peaks in mpSEs, (top left) duration of mpSEs. All axes have a minimum value of 0 and maximum value of 1. (**B**) Bar graph indicating proportion of the cluster-assignments for 4-day recordings in control (n=5) and PNA (n=4) animals. Data show mean ± SEM. (**C–D**) Representative examples of four-day consecutive recording of ARN^KISS activity and corresponding hourly k-means cluster assignments in (**C**) two control animals starting in proestrus on day 1 and transiting to diestrus on day 4 according to vaginal smears; (**D**) two PNA animals remaining in either metestrus or diestrus for four days. Light-off periods (19:00-07:00) are represented by the shaded area in the photometry recording. Color fields represent the assigned clusters for the center (4th hour) of the 7 hr moving time windows. Estrous stages of the mice determined by vaginal lavage at the beginning and the end of each 24 hr recording were labelled above the corresponding k-means cluster assignments: proestrus to estrus (PE), estrus to metestrus (EM), metestrus to diestrus (MD), stuck in metestrus (MM), or stuck in diestrus (DD). K-means clustering was performed without vaginal cytology information.

diestrous activity patterns. As expected (**McQuillan et al., 2019**), we found a marked suppression of ARN^KISS neuron SEs (n=3, progesterone main effect $F_{(1,10)}=7.333$, p=0.022) that lasted for up to 6 hr following progesterone in control mice (n=3, p<0.001, Sidak's multiple comparisons test, **Figure 7A, B and E**). The response in PNA mice was much less consistent, and overall, no significant differences in SE activity detected at any time point (n=3, progesterone main effect, $F_{(1,10)}=1.917$, p=0.196, **Figure 7C, D and F**).

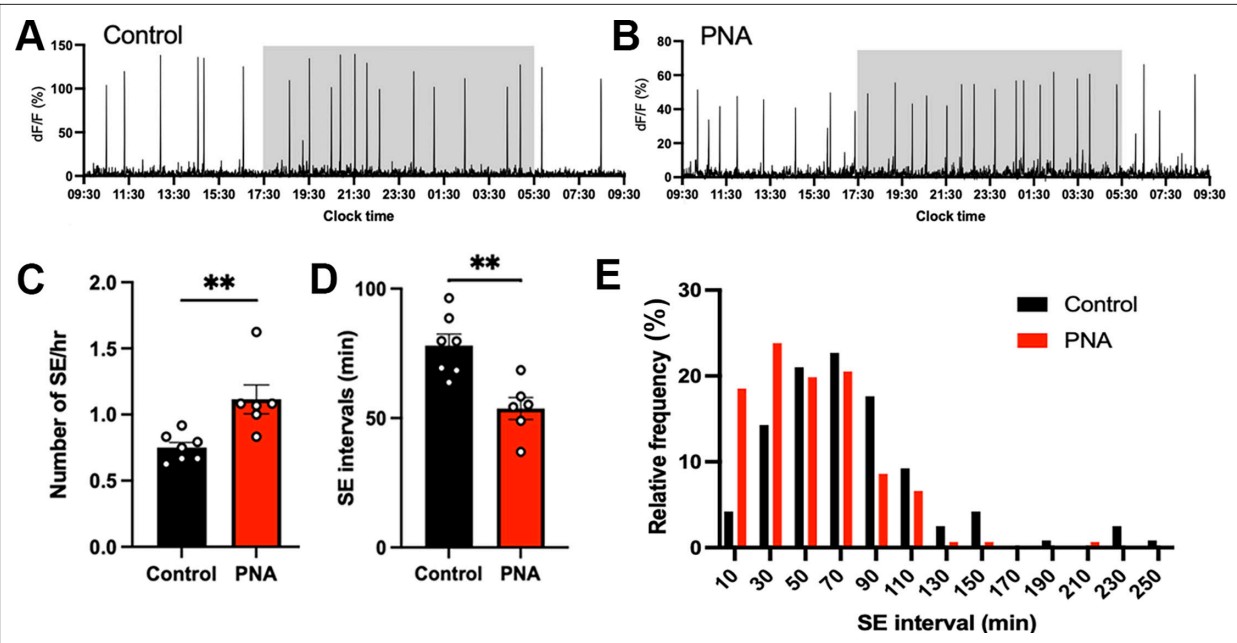

**Figure 6.** Faster ARN[KISS] neuron synchronization events in 'diestrous-like' PNA animals. Representative 24 hr photometry recordings showing ARN[KISS] neuron SEs observed in (**A**) diestrous control and (**B**) algorithm-identified diestrous PNA females. The light-off period (17:30–05:30) is represented by the shaded area. (**C**) SE frequency and (**D**) SE intervals in control (n=7) and PNA (n=6) mice. Mann-Whitney U tests. (**E**) Histogram showing percentage of SE interval frequencies occurring in 20 min bins, calculated separately for controls (black, n=119, 7 mice) and PNA (red, n=151, 6 mice) animals. X-axis represents the bin centers.

## No differences exist in ARN[KISS] neuron activity in ovariectomized (OVX) PNA and control females

Gonadal steroids play a key role in modulating the activity of the ARN[KISS] neurons (**McQuillan et al., 2019**; **McQuillan et al., 2022**). To assess whether any differences exist in ARN[KISS] neuron activity in PNA mice in the absence of gonadal steroids, we undertook 24 hr fiber photometry recordings in mice that had been ovariectomized for at least 3 weeks. Removal of gonadal steroids results in increased SE frequency and the emergence of clusters of tightly coupled individual synchronizations as previously described (**McQuillan et al., 2022**; **Figure 8A and B**). We observed no significant differences (Mann-Whitney U tests) between SE frequency (**Figure 8C**) or intervals (**Figure 8D and E**) in PNA and control animals (n=4 per group). Similarly, high-resolution recordings at 10 Hz did not reveal any differences in SE profiles between OVX control and OVX PNA mice (**Figure 8F and H**).

## Discussion

It is important to establish animal models that faithfully recapitulate the symptoms and features of women with PCOS to be able to make progress in treating the disorder (**Stener-Victorin et al., 2020**). Increased LH pulse frequency is observed in ~75% of women with PCOS and is considered to be a significant contributor to their sub-fertility (**Taylor et al., 1997**). We provide here a detailed assessment of the GnRH pulse generator in two commonly used mouse models of PCOS. The obese PPA mouse model was found to have a pulse generator that actually operates at a slower frequency than normal. In contrast, the PNA mouse model has a pulse generator that can operate somewhat faster than normal but, remarkably, maintains cyclical-like patterns of activity despite being in reproductively acyclic mice.

### Pulse generator activity in PPA mice

The chronic administration of DHT to female mice from postnatal day 21 generates an androgen receptor-dependent mouse model of PCOS with infertility and metabolic disturbances (**Caldwell et al., 2014**; **Caldwell et al., 2017**; **Stener-Victorin et al., 2020**; **Kerbus et al., 2024**). In agreement

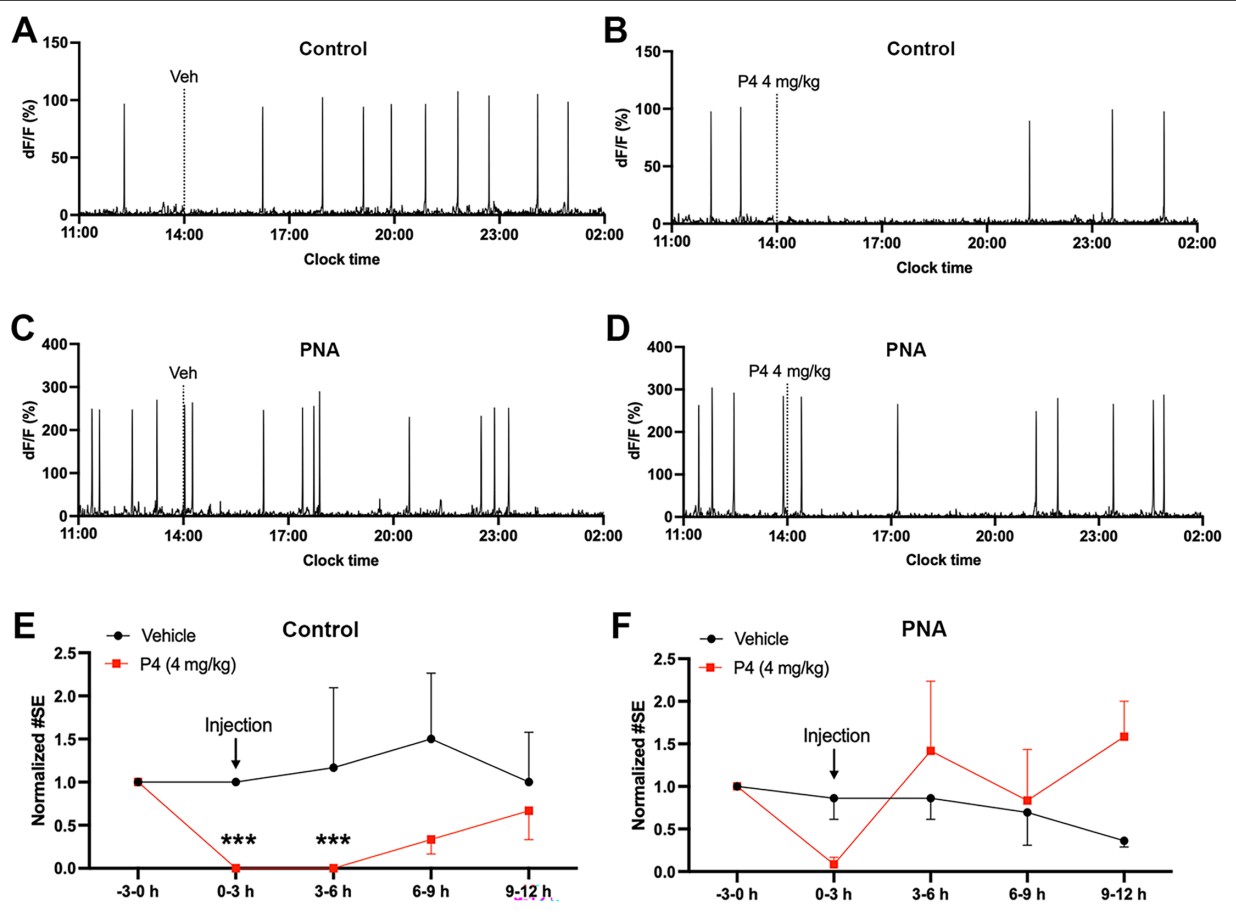

**Figure 7.** Defective progesterone negative feedback in PNA animals. (**A–D**) Examples of 15 hr photometry recordings of pulse generator activity in (**A–B**) control and (**C–D**) PNA animals receiving (**A,C**) vehicle (Veh) or (**B,D**) progesterone (P4, 4 mg/kg) i.p. injections. Dashed line represents when i.p. injection took place. Normalized number of SEs in (**E**) control (n=3) and (**F**) PNA (n=3) animals before and after vehicle (black) and P4 (red) injections. Data were analyzed in 3 hr bins and normalized to the number of SEs before injections (−3–0 hr) for each mouse. The asterisks above the data denote significant effects at indicated time periods compared to the pre-treatment period (−3–0 hr) by Sidak's multiple comparisons test. Data represent mean ± SEM. Two-way repeated-measure ANOVA with Sidak's multiple comparisons tests. *** p<0.001.

with prior studies, we found that PPA mice start to exhibit increased body weight around 4 weeks after DHT exposure and are acyclic. Fiber photometry revealed that the frequency of ARN$^{KISS}$ neuron SEs was reduced by ~50% in PPA mice compared to controls. This clearly indicates that the PPA mouse does not provide an appropriate PCOS model of elevated pulse generator activity. Given this, we did not proceed to examine pulsatile LH secretion in PPA mice but note that a recent study has reported that pulsatile LH secretion is not increased in this animal model (*Coyle et al., 2022*). It is very likely that chronic excess androgen activation in this model provides additional negative feedback suppression of ARN$^{KISS}$ neurons. The treatment of adult OVX female mice with the same DHT regimen is reported to decrease LH pulse frequency and amplitude (*Esparza et al., 2020*).

## Variable reproductive parameters of the PNA mouse model

The PNA mouse model has been studied by many different research groups with some features being identified in a highly consistent manner while others are not (*Stener-Victorin et al., 2020*). Common characteristics, also present in the current study, are those of increased anogenital distance, normal body weight, and highly disrupted estrous cycles (*Sullivan and Moenter, 2004*; *Silva et al., 2018*; *Stener-Victorin et al., 2020*; *Gibson et al., 2021*; *McCarthy et al., 2022*).

In contrast, reported differences in circulating gonadal steroid concentrations in PNA mice have been highly inconsistent among studies. Using ELISAs and radioimmunoassyays, some studies have demonstrated PNA mice to have elevated testosterone levels while others, including ourselves,

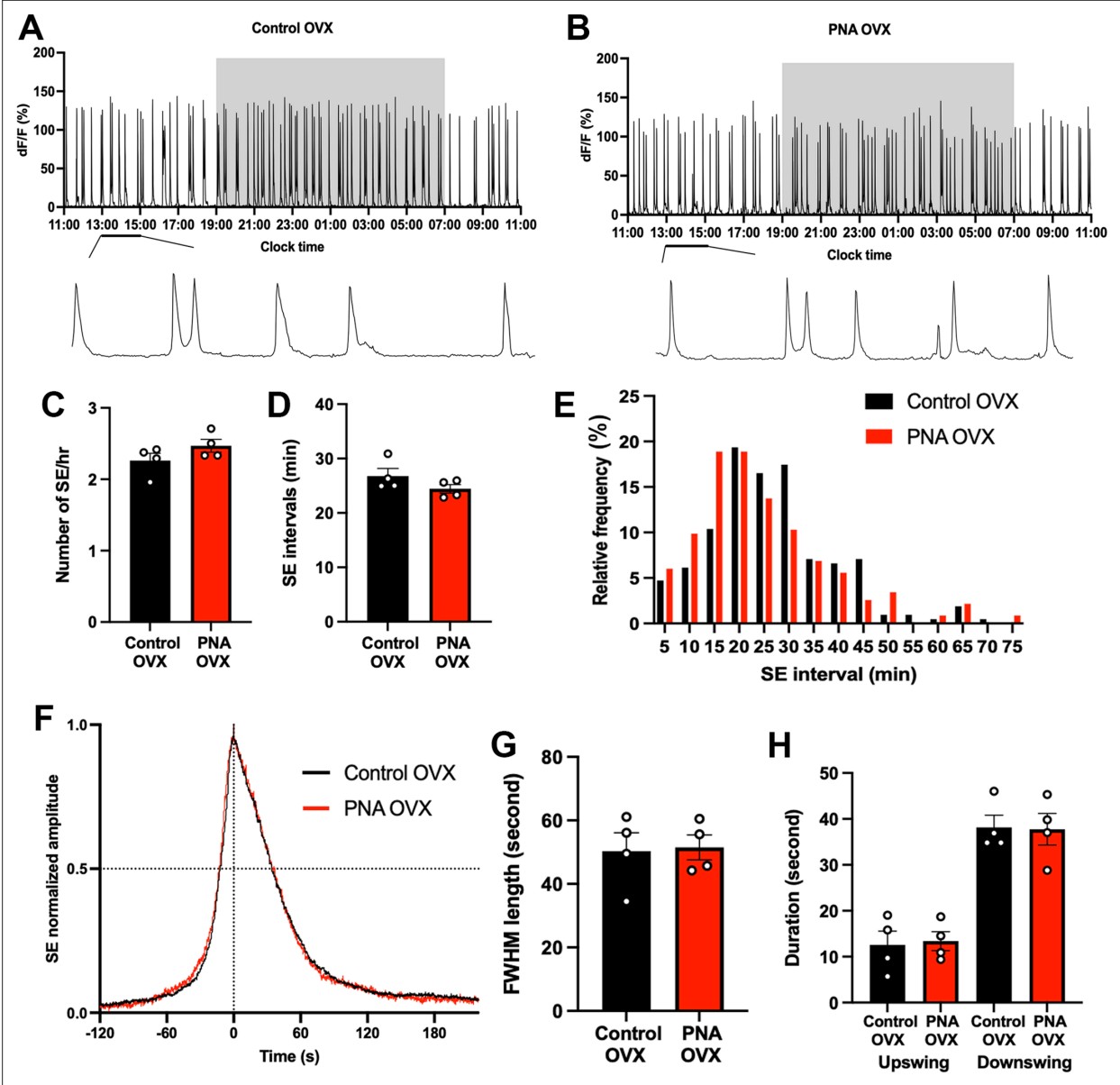

**Figure 8.** Similar ARN^KISS activity and SE profiles in control and PNA animals following ovariectomy. Representative 24 hr photometry recordings showing ARN^KISS neuron SEs observed in (**A**) control and (**B**) PNA females 3 weeks after ovariectomy (OVX) with the light-off period (19:00-07:00) represented by the shaded area and expanded views of the traces (13:00-15:00) given below (**A–B**). (**C**) SE frequency and (**D**) SE intervals in control and PNA mice (n=4), Mann-Whitney U tests. (**E**) Histogram showing relative SE frequencies occurring in 5 min bins, calculated separately for controls (n=212, 4 mice) and PNA (n=233, 4 mice) animals. X-axis represents the bin centers. Each SE cluster was analyzed as a single SE with the interval calculated from the first peak. (**F**) Continuous recordings at 10 Hz sampling rate showing normalized profile of SEs following OVX overlaid from control (black, 14 SEs from 4 animals) and PNA (red, 10 SEs from 4 animals). (**G**) FWHM length of OVX control and PNA animals (n=4). Mann-Whitney U test. (**H**) Durations of upswing and downswing for control and PNA animals (n=4), respectively. p>0.05, Kruskal-Wallis test followed by Dunn's multiple comparisons test. Data show mean ± SEM. Each circle is an individual animal.

find no differences (*Sullivan and Moenter, 2004*; *Roland et al., 2010*; *Roland and Moenter, 2011*; *Moore et al., 2015*; *Lei et al., 2017*; *Silva et al., 2018*). It is unclear why this is the case, but we note that some studies have demonstrated a dependence on age. For example, Roland and co-workers reported testosterone levels to be normal in 5-month PNA mice but elevated at 8 months (*Roland et al., 2010*) while Lei and colleagues found normal testosterone levels at ~1 month but elevated concentrations at 2–3 months (*Lei et al., 2017*). We assayed testosterone levels at 5–7 months of age in the present study.

Concerns over the specificity and selectivity of ELISAs for measuring gonadal steroids in mice (*Handelsman and Wartofsky, 2013*; *Auchus, 2014*; *Wierman et al., 2014*; *Handelsman et al., 2015*) led us to re-assess gonadal steroid hormone levels using gold-standard ultrasensitive LC-MS (*Handelsman et al., 2020*). We previously observed that testosterone levels were usually below the limit of detection in C57/BL6 female mice (*Wall et al., 2023*) but considered that any elevated concentrations in PNA mice might become detectable. Unfortunately, this was not the case, and overall it remains unclear if and when testosterone levels are elevated in PNA mice. Further detailed LC-MS assessment of testosterone levels will be required in the PNA model to address this important issue. We were, however, able to detect estradiol and progesterone concentrations and find that neither is different between control and PNA mice. A prior ELISA assessment also reported that estradiol and progesterone levels were unchanged in PNA animals (*Moore et al., 2015*).

Another area of considerable inconsistency among studies involves the effects of PNA treatment on both mRNA and protein expression of kisspeptin, neurokinin B, and dynorphin in the ARN (*Yan et al., 2014*; *Osuka et al., 2016*; *Gibson et al., 2021*; *McCarthy et al., 2022*; *Moore et al., 2021*). We re-investigated this question by quantifying the immunoreactive levels of these three proteins and observed an increase in only kisspeptin expression and only in rostral ARN of PNA animals. While there are important roles for these neuropeptides in modulating ARN$^{KISS}$ neuron synchronizations (*Han et al., 2023*; *Morris and Herbison, 2023*) and their subsequent activation of GnRH secretion (*Liu et al., 2021*), these generally inconsistent findings indicate that any PNA-induced changes in the expression of the neuropeptides are subtle. Indeed, the principal determinant of kisspeptin, neurokinin B, and dynorphin expression in ARN$^{KISS}$ neurons is estradiol (*Navarro et al., 2009*) and we demonstrate here that circulating estradiol levels are not altered in PNA mice.

## Pulse generator activity and pulsatile LH in the PNA mouse model

One of the key features of the PNA mouse model is thought to be that of increased pulsatile LH secretion (*Stener-Victorin et al., 2020*). However, this has only been reported in a single study with PNA mice exhibiting an approximately 40% increase in LH pulse frequency (*Moore et al., 2015*). A recent paper undertaking the same tail-tip pulse bleeding approach found that LH pulse frequency was not significantly altered in PNA mice (*McCarthy et al., 2022*). Our present observations may explain these discrepancies. As the pulse generator continues to cycle in the acyclic PNA mouse, it is essentially impossible to define the 'pulse generator stage' of a PNA mouse at the time of bleeding and consequently wide variations in LH pulse frequency would be expected. We find here that LH pulse frequency recorded from such random 'pulse generator' cycling PNA mice is not significantly different to that of controls. Nevertheless, with the benefit of being able to measure ARN$^{KISS}$ neuron activity directly, it has been possible to show that 'diestrous-stage' PNA mice do have an increased rate of pulse generator frequency compared with controls. This may account for the overall increase in mean LH levels detected in the present study. As such, the PNA mouse model may to some degree replicate the increased pulse frequency seen in many women with PCOS.

The most striking feature of pulse generator activity in PNA mice is its ability to retain cyclical patterns of activity within a reproductively acyclic mouse. The pulse generator of PNA mice exhibits clear transitions from periods of 'estrous-like' Cluster_0 activity through to mpSE-dominated Cluster_3/4 phase and onto regular 'diestrous-like' Cluster_1 and _2 patterns. Although the patterns of activity were not always identical, the overall percentage of time PNA mice spend in each pattern was not found to be significantly different to controls. We also note that the dynamics of individual ARN$^{KISS}$ neuron SEs remain unchanged under all conditions in PNA mice, suggesting that the fundamental synchronization mechanism and numbers of contributing neurons is not altered (*Han et al., 2023*). This is consistent with the observation that PNA treatment does not alter ARN$^{KISS}$ neuron firing activity in adult acute brain slices (*Gibson et al., 2021*). Together, these observations indicate that the acyclic reproductive phenotype of PNA mice does not result from an absence of cyclical activity in the pulse generator. Although pulse generation in the mouse does not depend on action potential generation by GnRH neurons (*Wang et al., 2020*; *Herbison, 2021*), it is noteworthy that a significant impact of PNA treatment on the activity of the GnRH neuron can be excluded (*Jaime and Moenter, 2022*). The degree to which cyclical GnRH pulse generator activity in PNA mice reflects women with PCOS is unknown. However, a very wide range of LH pulse frequency is found in anovulatory women with PCOS (*Taylor et al., 1997*), suggesting

the intriguing possibility that the GnRH pulse generator activity may also continue to cycle in women with PCOS.

## Gonadal steroid feedback in PNA mice

Pulse generator activity was found to be identical in ovariectomized PNA and control mice. This indicates that, despite androgen receptors being expressed by ARN^KISS neurons in the perinatal period (*Watanabe et al., 2023*), excess DHT exposure at this time does not engender any fundamental differences in their synchronization behavior in the absence of gonads as adults. This suggests that the subtle increase in pulse generator frequency detected here in 'diestrous-like' PNA mice arises from the effects of circulating gonadal steroids. As gonadal steroid levels are normal in PNA mice, it is likely that perinatal DHT exposure programs long-term changes in gonadal steroid receptor-dependent effectors in adult PNA mice. Direct estrogen feedback is critical in maintaining the suppressed activity of the pulse generator throughout the estrous cycle (*McQuillan et al., 2022*), but ARN^KISS neuron ESR1 expression was not found to be altered in PNA mice (*Moore et al., 2021*).

Reduced progesterone negative feedback is widely considered to be a major factor underlying enhanced LH pulsatility in women with PCOS (*Pastor et al., 1998*; *Chhabra et al., 2005*) and the ability of progesterone to suppress LH secretion is reduced in OVX PNA mice (*Moore et al., 2015*). Whereas progesterone exerts a robust and prolonged suppression of ARN^KISS neuron SEs in normal (*McQuillan et al., 2019*) and control mice, we find this to be much less effective in PNA females. Precisely how perinatal DHT treatment desensitizes ARN^KISS neurons to progesterone is unknown and could occur directly or indirectly (*Moore et al., 2021*). Reported effects of PNA on progesterone receptors in the ARN are inconsistent with studies finding either reduced or unchanged expression (*Moore et al., 2015*; *Gibson et al., 2021*; *Moore et al., 2021*). For example, one recent study found a 30% reduction in progesterone receptor mRNA expression by ARN^KISS neuron in PNA mice (*Moore et al., 2021*).

Although we observed that the PNA mouse has reduced progesterone feedback at the level of the pulse generator, as is suspected for women with PCOS, it is unclear how this might pattern ARN^KISS neuron activity. Control and PNA mice exhibit similar proportions of time with halted or slow pulse generator activity (Cluster_0). This pattern of activity is thought to occur in response to increased progesterone level following the LH surge (*McQuillan et al., 2019*; *Vas et al., 2024*). Thus, it is somewhat perplexing to find clear Cluster_0 activity in PNA mice that have reduced sensitivity to progesterone negative feedback. This might suggest that other mechanisms can become responsible for luteal phase slowing of pulse generator activity in PNA mice. Equally, is unclear how reduced progesterone feedback could be responsible for elevated 'diestrous-like' patterns of ARN^KISS neuron activity, although roles for progesterone outside the luteal phase have been suggested (*Leipheimer et al., 1984*).

Previous studies have demonstrated that the estrogen-activated LH surge mechanism is normal in PNA mice (*Moore et al., 2013*) and, despite proestrous smears never being observed in PNA mice, copra lutea can be found indicating that LH surges and ovulation can occur (*Moore et al., 2013*; *Lei et al., 2017*; *McCarthy et al., 2022*). This raises the intriguing possibility that the GnRH surge generator may also be operational in PNA mice but masked by abnormal downstream signalling. In that regard, we note that pituitary gonadotroph sensitivity to GnRH is blunted in PNA mice as previously reported (*Silva et al., 2019*). As pulse generator activity is the same in ovariectomized PNA and control mice, reduced gonadotroph sensitivity might explain the reduced rise in LH following ovariectomy in PNA mice observed in other studies (*Moore et al., 2013*; *Moore et al., 2015*). Nevertheless, we note that the PNA mouse clearly does not model women with PCOS that show a much-elevated LH response to GnRH (*Barnes et al., 1989*; *Batrinos, 1993*; *Morales et al., 1996*, *Patel et al., 2003*).

In summary, we report here that the GnRH pulse generator operates at a lower frequency in PPA mice, but at a higher frequency in PNA mice when 'diestrous-like' pulse generator activity was analyzed. However, we note that many features of PNA mice such as their pulse generator free-running behavior in the ovariectomized state and gonadal steroid hormone concentrations are normal. Remarkably, we find evidence that the pulse generator continues to exhibit estrous cycle-like patterns of activity within an acyclic PNA mouse model. Thus, the anovulatory phenotype of PNA mice likely arises in a complex manner from deficits at many levels of the hypothalamo-pituitary-ovarian axis. The degree

to which the PNA mouse model reflects women with PCOS remains an important, albeit very difficult, question to address.

## Methods

### Mice

Female 129S6Sv/Ev C57BL/6 *Kiss1*[Cre/+] mice (*Yeo et al., 2016*) were crossed with the Ai162 (TIT2L-GC6s-ICL-tTA2)-D Cre-dependent GCaMP6s males (JAX stock #031562; *Daigle et al., 2018*) to generate *Kiss1*-Cre/+,*Ai126D* mice as previously characterized (*Han et al., 2023*). Mice were group-housed in conventional cages with environmental enrichment under conditions of controlled temperature (22 ± 2 °C) and lighting (12 hr light/12 hr dark cycle; lights on at 05:30 or 07:00) with ad libitum access to food (RM3 (E), Special Diets Services, UK) and water. All animal experimental protocols were approved by the University of Cambridge, UK (P174441DE). Where required, mice were bilaterally ovariectomized (OVX) under isoflurane anesthesia at least 3 weeks prior to experimentation.

### Peripubertal androgen (PPA) model

Peripubertal female mice were exposed to DHT as previously described (*van Houten et al., 2012*; *Caldwell et al., 2017*). Briefly, peripubertal female mice (21–23 days old) were implanted subcutaneously (s.c.) with either an empty (control) or dihydrotestosterone (DHT)-filled 1 cm silastic implants (id, 1.47 mm; od, 1.95 mm, Dow Corning Corp, catalog no. 508–006). These silastic implants containing ~10 mg DHT provide steady-state DHT release for at least 6 months (*Singh et al., 1995*). The body weight of these animals was monitored weekly until optic fiber implantation at the 10 weeks of age.

### Prenatal androgen (PNA) model

Adult females were paired with males and checked for copulatory plugs, indicating day 1 of gestation. Pregnant dams were given s.c. injections of sesame oil vehicle (100 µL) alone or containing 250 µg of DHT on gestational days 16, 17, and 18 (*Sullivan and Moenter, 2004*). Female offspring from androgen-treated (PNA) and vehicle-treated (control) dams were used for experiments.

### Estrous cycle determination

The estrous cycle stage was determined by the relative ratio of leukocytes, cornified, and nucleated cells observed in the vaginal epithelial cell smear samples on the morning of collection (*Ajayi and Akhigbe, 2020*). The smears were collected using 5 µL of sterile phosphate-buffered saline (PBS) which was transferred to a glass slide to air dry before staining with filtered Giemsa (1:1 in MilliQ water) and examined under a light microscope.

### Testosterone ELISA measurements and analysis

Blood samples for testosterone measurement were collected in the afternoon of diestrus from 22- to 30-week-old control and PNA animals through a tail-tip blood sample prior to ovariectomy or from the inferior vena cava before perfusion. The samples were then spun at room temperature at 3000 rpm for 10 min. Serum was collected and kept at –20 °C until the day of hormone measurement. Serum testosterone levels were measured using the testosterone rat/mouse ELISA kit according to the manufacturer's instructions (Demeditec Diagnostics, GmbH, DEV9911, Kiel, Bermany). The assay sensitivity for mouse testosterone ELISA was 0.066 ng/mL, and intra-assay coefficient of variation was 6.5%.

### Liquid chromatography–Mass spectrometry (LC-MS)

Terminal blood samples were collected from the inferior vena cava of anaesthetized 18- to 22-week-old control and PNA animals at 10:00 on diestrus exactly as described previously (*Wall et al., 2023*). Plasma levels of estradiol, progesterone, testosterone, and androstenedione were measured using liquid chromatography-mass spectrometry (LC-MS) with detection limits of androstenedione: 0.05 ng/mL; estradiol: 0.50 pg/mL; progesterone: 0.05 ng/mL; testosterone: 0.01 ng/mL (*Handelsman et al., 2020*).

## Quantification of kisspeptin/neurokinin B/dynorphin expression in ARN

Adult diestrous PNA and control female mice were given a lethal dose of pentobarbital (3 mg/100 μL, intraperitoneal, i.p.) and perfused transcardially with 4% paraformaldehyde (PFA). Brains were post-fixed in 4% PFA for 1 hr at 4 °C and transferred to 30% sucrose at 4 °C until sunk. Forty-μm-thick sections were collected with microtomes and processed for kisspeptin and neurokinin B (NKB) immunofluorescence and dynorphin (DYN) Nickel-3,3'-Diaminobenzidine (DAB) staining. For kisspeptin staining, brain sections were incubated with rabbit anti-kisspeptin (1:5000, AC566; a gift from Dr. Alain Caraty) followed by goat anti-rabbit Alexa Fluor 568 (1:500, Molecular Probes). For NKB staining, brain sections were incubated with guinea pig anti-NKB (1:5000, IS-3/61, a gift from Dr. Philippe Ciofi) followed by biotinylated goat anti-guinea pig IgG (1:1000, Vector Laboratories) and DyLight Streptavidin 488 (1:500, BioLegend). Images were taken using a Leica SP8 Laser Scanning Confocal Microscope (Leica Microsystems) at the Cambridge Advanced Imaging Center and analyzed using ImageJ. Brain sections for DYN Nickel DAB labeling were incubated with rabbit anti-DYN B (1:12,000, a gift from Dr. Philippe Ciofi; *Griffond et al., 1993* followed by biotinylated goat anti-rabbit IgG (1:200, Vector Laboratories)). Brain sections were further incubated in avidin-biotin peroxidase solution before reacting with Nickel DAB developing solution. Images were taken using an Olympus BX43 upright microscope with DP74 digital camera (Olympus Life Science) and analyzed using Image J.

## Stereotaxic surgery and GCaMP fiber photometry

Adult female PNA, PPA, and their respective control *Kiss1*-Cre/+,*Ai126D* mice (10–14 weeks old) were anesthetized with 2% isoflurane, given meloxicam (5 mg/kg, s.c.), buprenorphine (0.05 mg/kg, s.c.), and dexamethasone (10 mg/kg, s.c.) and placed in a stereotaxic apparatus. A small hole was drilled in the skull and a unilateral indwelling optical fiber (400 μm diameter; 0.48 NA, Doric LenSEs, Quebec, Canada) was implanted directly above the mid-caudal ARN (2.0 mm posterior to bregma, 3.0 mm lateral to the superior sagittal sinus, 5.93 mm deep). Following optic fiber implantation, mice were monitored through recovery for 5 days and received post-operative analgesics (meloxicam, 5 mg/kg; orally) for up to 2 days. Then mice were handled daily for at least 3 weeks and habituated to the fiber photometry recording setup before fluorescence signals were recorded from freely behaving mice in their home cages.

Fiber photometry was undertaken as detailed previously (*Clarkson et al., 2017*). Photometry systems were built using Doric components (Doric Lenses, QC, Canada) and National Instrument data acquisition board (TX, USA) based on a previous design (*Lerner et al., 2015*). Blue (465–490 nm) and violet (405 nm) LED lights were sinusoidally modulated at frequencies of 531 and 211 Hz, respectively. Both lights were focused onto a single fiber optic connected to the mouse. The light intensity at the tip of the fiber was 30–80 micro watts. Emitted fluorescence signal from the brain was collected via the same fiber, passed through a 500–550 nm emission filter, before being focused onto a fluorescence detector (Doric, QC, Canada). The emissions were collected at 10 Hz and the two GCaMP6 emissions were recovered by demodulating the 465–490 nm signals (calcium-dependent) and 405 nm (calcium independent) signals.

Fluorescence signals were sampled using either a scheduled mode (5/15 s on/off) or a continuous mode of light emission with custom software (Tussock Innovation). Twenty-four-hour recordings were made from PNA, PPA, and their respective control female mice in the diestrous stage starting 4 hr after lights-on. Four-day recordings were started on proestrus for control mice and on any diestrous or metestrous stage for PNA mice. Continuous data acquisition mode was used to provide high temporal resolution comparisons of SEs. Signals from each SE were normalized to its peak (intact: –120 to +120 s; OVX: –120 to +200 s from the peak time). The full widths at half maximum (FHWM) of SE were determined at half-maximal amplitude of the normalized SEs. Averaged SE parameters for each animal were calculated before computing the means for each treatment group.

Analysis was performed in MATLAB with the subtraction of 405 signal from 465-490 signal to extract the calcium-dependent fluorescence signal. An exponential fit algorithm was used to correct for baseline shift. The signal was calculated in dF/F (%) with the equation $dF/F = (F_{fluorescence}-F_{baseline})/(F_{baseline}) \times 100$. The Findpeaks algorithm was used to detect SEs with peaks in dF/F greater than 50% of the maximum signal in each recording. Inter-SE intervals represent the time between peak values of each SEs. Multi-peak SE in PNA animals and cluster SE in OVX were analyzed as one SE, and SE intervals were calculated from the first peak.

## Serial blood sampling, GnRH injection, and LH measurement

Mice were handled daily for at least 3 weeks to habituate for tail-tip bleeding. To examine the relationship between ARN$^{KISS}$ neuron SEs and pulsatile LH secretion, freely behaving mice were attached to the fiber photometry system, and 4 µL blood samples were taken every 6 or 12 min from the tail-tip over a period of 2–4 hr during recording. For standard serial blood sampling experiments, LH secretion was measured by obtaining 4 µL blood samples every 10 min for 4 hr (between 12:00 and 17:00). At the end of the bleeding period, 200 ng/kg body weight of GnRH (Bachem, catalog No. 4033013, Switzerland) was injected i.p., and two more blood samples were collected at 10 min intervals. Whole blood samples were diluted 1:15 in 1 x PBS with 0.05% Tween 20 (PBS-T) for LH ELISA (*Steyn et al., 2013*) with an assay sensitivity of 0.04 ng/mL, and intra-assay and interassay coefficients of variation of 9.3% and 10.5%, respectively. Pulses were defined and analyzed by PULSAR with the settings optimized for intact female mice: G1=3.5, G2=2.6, G3=1.9, G4=1.5, and G5=1.2 (*Porteous et al., 2021*).

## K-means clustering

Data analysis and k-means clustering analysis were performed without vaginal cytology information as previously described (*Vas et al., 2024*). Synchronization events with multiple peaks occurring within 160 s were defined as mpSEs. In brief, 4 day recordings were segmented into 24 hr periods before applying a customized MATLAB code to extract features of the population activity of the ARN$^{KISS}$ neurons including the (i) number of SEs within a given time period; (ii) prevalence of different type of SEs (single- and mpSEs); (iii) inter-SE intervals, variation in the inter-SE intervals (reflecting regularity of activity patterns); and (iv) shape of the SEs (height, variation in height, half-duration). These 24 hr periods were further fragmented using a 7 hr window size with 1 hr sliding steps. The first and the last 3 hr of each 24 hr recording were extrapolated from the assignment of the 4th and 21st hr, respectively. The following parameters were used to describe the activity pattern of ARN$^{KISS}$ neurons: (i) number of SEs, where mpSEs were counted as a single SE; (ii) standard deviation (SD) of inter-SE intervals; (iii) number of mpSEs; (iv) average number of peaks within an mpSEs; and (v) average duration of mpSEs. All these parameter values were then linearly rescaled to values between 0 and 1. To perform k-means clustering on the data set, we used RapidMiner Studio's (https://rapidminer.com, educational license) built-in module (k=5, number of runs with random initiations = 100, measure types = 'NumericalMeasures', numerical measure = 'EuclideanDistance', max optimization steps = 100). Cluster centroid data and cluster labels (to each 7 hr time window) were exported from RapidMiner Studio and were visualized in R (using 'ggplot2', 'readr' and 'fmsb' libraries).

## Progesterone injection

Progesterone stock solution (800 µg/mL) was prepared by dissolving 8 mg of progesterone (Sigma, Catalog No. P0130) in 1 mL ethanol followed by 1:10 dilution in sesame oil. Diestrous female mice were recorded for 3 hr before an i.p. injection of progesterone (4 mg/kg body weight) or vehicle (100 µL, sesame oil). A previous study has shown that i.p. injection of progesterone at 8 mg/kg body weight to C57BL/6 mice results in peak circulating progesterone concentrations of 110 ng/mL before dropping to 34 ng/mL 1 hr later (*Wong et al., 2012*). As circulating progesterone levels at proestrus in C57BL/6 mice are 22±4 ng/mL (*Wall et al., 2023*), we employed half of that dose to test the sensitivity of progesterone negative feedback to ARN$^{KISS}$ neurons.

## Statistical analysis

All statistical analyses were performed using GraphPad Prism 10 software. Depending on different experimental designs, and the nature of data distribution, data were analyzed by Mann-Whitney U tests, Kruskal-Wallis test followed by Dunn's multiple comparison tests, two-way ANOVA followed by Sidak's multiple comparisons test, or two-way repeated-measures ANOVA followed by Sidak's multiple comparisons test. Samples sizes were based on prior experiments. Where possible investigators were blind to the experimental groupings. The threshold level for statistical significance was set at $p < 0.05$ with data presented as mean ± SEM with each dot representing one animal.

## Acknowledgements

This research was supported by the Wellcome Trust (212242/Z/18/Z) to AEH and Medical Research Council (MR N013433-1) and Harding Distinguished Postgraduate Scholars Programme Leverage Scheme to ZZ.

## Additional information

### Funding

| Funder | Grant reference number | Author |
| --- | --- | --- |
| Wellcome Trust | 10.35802/212242 | Allan E Herbison |
| Medical Research Council | MR N013433-1 | Ziyue Zhou |

The funders had no role in study design, data collection and interpretation, or the decision to submit the work for publication. For the purpose of Open Access, the authors have applied a CC BY public copyright license to any Author Accepted Manuscript version arising from this submission.

### Author contributions

Ziyue Zhou, Formal analysis, Investigation, Methodology, Writing – original draft, Writing – review and editing; Su Young Han, Supervision, Methodology, Writing – review and editing; Maria Pardo-Navarro, Ellen G Wall, Investigation, Writing – review and editing; Reena Desai, Investigation; Szilvia Vas, Investigation, Methodology, Writing – review and editing; David J Handelsman, Resources, Supervision, Writing – review and editing; Allan E Herbison, Conceptualization, Supervision, Funding acquisition, Writing – original draft, Project administration, Writing – review and editing

### Author ORCIDs

Ziyue Zhou ⓘ https://orcid.org/0000-0003-4725-7543
Allan E Herbison ⓘ https://orcid.org/0000-0002-9615-3022

### Ethics

All animal handling and experimental protocols were undertaken as approved by the Animal Welfare and Ethical Review Body of the University of Cambridge under UK Home Office project license P174441DE. All surgery was performed under isofluorane anesthesia, and every effort was made to minimize suffering.

Reviewer #2 (Public review): https://doi.org/10.7554/eLife.97179.3.sa1
Reviewer #3 (Public review): https://doi.org/10.7554/eLife.97179.3.sa2
Author response https://doi.org/10.7554/eLife.97179.3.sa3

## Additional files

### Supplementary files

MDAR checklist

### Data availability

All data analyzed during this study are included in the manuscript and source data for figures have been uploaded to Dryad https://doi.org/10.5061/dryad.mpg4f4r97.

The following dataset was generated:

| Author(s) | Year | Dataset title | Dataset URL | Database and Identifier |
|---|---|---|---|---|
| Zhou Z, Han SY, Pardo-Navarro M, Wall EG, Desai R, Vas S, Handelsman D, Herbison AE | 2024 | GnRH pulse generator activity in mouse models of polycystic ovary syndrome | https://doi.org/10.5061/dryad.mpg4f4r97 | Dryad Digital Repository, 10.5061/dryad.mpg4f4r97 |

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
