## [Editor Report · eLife Assessment]

This **important** study reports findings on the GnRH pulse generator's role in androgen-exposed mouse models, providing further insights into PCOS pathophysiology and advancing the field of reproductive endocrinology. The experimental data were collected using cutting-edge methodologies and are **solid**. The findings, while interesting, are primarily applicable to mouse models, and their translation to human physiology requires cautious interpretation and further validation. This work will be of interest to endocrinologists and reproductive biologists.

---

## [Referee Report · Reviewer #2 (Public review)]

Summary:

The authors aimed to investigate the functionality of the GnRH (gonadotropin-releasing hormone) pulse generator in different mouse models to understand its role in reproductive physiology and its implications for conditions like polycystic ovary syndrome (PCOS). They compared the GnRH pulse generator activity in control mice, peripubertal androgen (PPA) treated mice, and prenatal androgen (PNA) exposed mice. The study sought to elucidate how androgen exposure affects the GnRH pulse generator and subsequent LH (luteinizing hormone) secretion, contributing to the pathophysiology of PCOS.

Strengths:

(1) Comprehensive Model Selection: The use of both PPA and PNA mouse models allows for a comparative analysis that can distinguish the effects of different timings of androgen exposure.

(2) Detailed Methodology: The methods employed, such as photometry recordings and serial blood sampling, are robust and allow for precise measurement of GnRH pulse generator activity and LH secretion.

(3) Clear Results Presentation: The experimental results are well-documented with appropriate statistical analyses, ensuring the findings are reliable and reproducible.

(4) Relevance to PCOS: The study addresses a significant gap in understanding the neuroendocrine mechanisms underlying PCOS, making the findings relevant to both basic science and potentially clinical research.

Weaknesses

(1) Model Limitations: While the PNA mouse model is suggested as the most appropriate for studying PCOS, the authors acknowledge that it does not completely replicate the human condition, particularly the elevated LH response seen in women with PCOS.

(2) Complex Data Interpretation: The reduced progesterone feedback and its effects on the GnRH pulse generator in PNA mice add complexity to data interpretation, making it challenging to draw straightforward conclusions.

(3) Machine Learning (ML) Selection and Validation: While k-means clustering is a useful tool for pattern recognition, the manuscript lacks detailed justification for choosing this specific algorithm over other potential methods. The robustness of clustering results has not been validated.

(4) Biological Interpretability: Although the machine learning approach identified cyclical patterns, the biological interpretation of these clusters in the context of PCOS is not thoroughly discussed. A deeper exploration of how these clusters correlate with physiological and pathological states could enhance the study's impact.

(5) Sample Size: The study uses a relatively small number of animals (n=4-7 per group), which may limit the generalisability of the findings. Larger sample sizes could provide more robust and statistically significant results.

(6) Scope of Application: The findings, while interesting, are primarily applicable to mouse models. The translation to human physiology requires cautious interpretation and further validation.

Comments on revised version:

I did not find the response to my main concerns regarding justification for the choice of the number of clusters (k) and providing evidence of cluster robustness satisfactory at all. It sounds contradictory to me to state that the authors have used unsupervised ML approach when at the same time had clear understanding of the data and the features they wanted to capture. Unsupervised approaches are meant to reveal features that are not apparent by eye... however in their response the authors state, "...our aim was to develop an unsupervised approach that would automatically detect the onset and existence of the key features of pulse generator cyclicity that were apparent by eye...". This sounds like a rather supervised ML approach to me.

Furthermore, I am still unsure why did the authors choose k=5, i.e. assumed there are 5 clusters in the data, and did they explore other possible values for k?

- If not why not? How does this fit with the claims that their ML approach is unsupervised, in other words purely data-driven without making any assumptions?

- If yes did they compare the robustness of their clustering results obtained for different values of k?

---

## [Referee Report · Reviewer #3 (Public review)]

Summary:

Zhou and colleagues elegantly used pre-clinical mouse models to understand the nature of abnormally high GnRH/LH pulse secretion in polycystic ovary syndrome (PCOS), a major endocrine disorder affecting female fertility worldwide. This work brings a fundamental question of how altered gonadotropin secretion takes place upstream within the GnRH pulse generator core, which is defined by arcuate nucleus kisspeptin neurons.

Strengths:

Authors use state-of-the-art in vivo calcium imaging with fiber photometry and important physiological manipulations and measurements to dissect the possible neuronal mechanisms underlying such neuroendocrine derangements in PCOS. The additional use of unsupervised k-means clustering analysis for the evaluation of calcium synchronous events greatly enhances the quality of their evidence. The authors nicely propose that neuroendocrine dysfunction in PCOS might involve different setpoints through the hypothalamic-pituitary-gonadal (HPG) axis, and beyond kisspeptin neurons, which importantly pushes our field forward toward future investigations.

Weaknesses:

The reviewer agrees that the authors provide important evidence and have improved the quality of the manuscript following first-round revisions. However, they seem resistant to show frequency and amplitude averages in Figure 1 or as supplemental data. Whether the amplitude is dependent on fiber position and its influences on the analysis should be a point of discussion and not data omission. A more detailed analysis of frequency data would enhance the quality of their manuscript.

Comments on revised version:

This comment is related to Reviewer 3's comment # 2 (major) response:

The response does not justify why authors could simply show frequency and amplitude averages in Figure 1 or as supplemental data. Whether the amplitude is dependent on fiber position and its influences on the analysis should be a point of discussion and not data omission.

---

## [Author Response]

The following is the authors’ response to the previous reviews.

**Reviewer #1 (Public Review):**
The manuscript involves 11 research vignettes that interrogate key aspects of GnRH pulse generator in two established mouse models of PCOS (peripubertal and prenatal androgenisation; PPA and PNA) (9 of the vignettes focus on the latter model).A key message of this paper is that the oft-quoted idea of rapid GnRH/LH pulses associated with PCOS is in fact not readily demonstrable in PNA and PPA mice. This is an important message to make known, but when established dogmas are being challenged, the experiments behind them need to be robust. In this case, underpowered experiments and one or two other issues greatly limit the overall robustness of the study.General critiques(1) My main concern is that many/most of the experiments were limited to 4-5 mice per group (PPA experiments 1 and 2, PNA experiments 3, 5, 6, 8, and 9). This seems very underpowered for trying to disprove established dogmas (sometimes falling back on "non-significant trends" - lines 105 and 239).

For the key characterization of GnRH pulse generator activity and LH pulsatility in intact PNA mice (Fig.3, 4, 6), we used 6-8 animals in each experiment which we believe to be sufficient.

It is pertinent to explore the “established dogma”. While there is every expectation that the PNA model should have increased LH pulsatility, in fact there is only a single study (Moore, Prescott et al. 2015) that has shown this. The two other reports that have examined this issue find no change in LH pulse frequency (McCarthy, Dischino et al. 2021 and ours). Hence, we would suggest that expectations rather than evidence presently maintains the PNA “dogma”. For the PPA model, there is in fact not a single paper reporting increased LH pulse frequency.

(2) Page 133-142: it is concerning that the PNA mice didn't have elevated testosterone levels, and this clearly isn't the fault of the assay as this was re-tested in the laboratory of Prof Handelsman, an expert in the field, using LCMS. The point (clearly made in lines 315-336 of the Discussion) that elevated testosterone in PNA mice has been shown in some but not other publications is an important concern to describe for the field. However, the fact remains that it IS elevated in numerous studies, and in the current study it is not so, yet the authors go on to present GnRH pulse generator data as characteristic of the PNA model. Perhaps a demonstration of elevated testosterone levels (by LCMS?) should become a standard model validation prerequisite for publishing any PNA model data.

We provide a Table below showing the huge inconsistencies in testosterone levels reported in the PNA mouse model. If anything, these inconsistencies might be explained by age, although again this is very variable between studies. Much the same as the “dogma” related to LH pulsatility in the PNA model, we would question whether there is any robust increase in testosterone levels in this model. There is no question that women with PCOS have elevated testosterone but whether the PNA mouse is a good model for this is debatable. We have noted this caution and the need for further LC-MS studies in the Discussion.

**Author response table 1. sa3table1:** 

Testosterone levels	Age of mice	Measurement	Source
Increased in PNA	4-6 months	Radioimmunoassay (Diagnostic Products, Los Angeles)	Sullivan and Moenter 2004
	8 months	Radioimmunoassay (Millipore, cat# SRI13K)	Roland,Nunemaker et al. 2010
	60-80 days ("∼2-3)
months"	ELISA (Demeditec Diagnostics, GmnH)^(**)	Moore, Prescott etal. 2015
	2 & 3 months	ELISA (YANYU,Shanghai, China)	Lei, Ding et al. 2017
	50-60 days (1.7-2months)	ELISA (LDN)	Silva, Prescott etal. 2018
No difference	5 months	Radioimmunoassay (Millipore, cat# SRI13K)	Roland,Nunemaker et al. 2010
	2-4 months	Radioimmunoassay (catalog no. TKTT2; Siemens Medical Solutions, Los Angeles, CA)	Roland and Moenter 2011
	3 weeks (1 month)	ELISA(YANYU,Shanghai, China)	Lei, Ding et al. 2017
	30-40 days (1-1.3 months)	ELISA (LDN)	Silva, Prescott et al. 2018

*Same ELISA used in the current study.

(3) Line 191-196: the lack of a significant increase in LH pulse frequency in PNA mice is based on measurements using reasonable group sizes (7-8), although the sampling frequency is low for this type of analysis (10-minute intervals; 6-minute intervals would seem safer for not missing some pulses). The significance of the LH pulse frequency results is not stated (looks like about p=0.01). The authors note that LH concentration IS elevated (approximately doubled), and this clearly is not caused by an increase in amplitude (Figure 4 G, H, I). These things are worth commenting on in the discussion.

We have included the p-value of the LH pulse frequency results and included the relevant discussion.

(4) An interesting observation is that PNA mice appear to continue to have cyclical patterns of GnRH pulse generator activity despite reproductive acyclicity as determined by vaginal cytology (lines 209-241). This finding was used to analyse the frequency of GnRH pulse generator SEs in the machine-learning-identified diestrous-like stage of PNA mice and compare it to diestrous control mice (as identified by vaginal cytology?) (lines 245-254). The idea of a cycle stage-specific comparison is good, but surely the only valid comparison would be to use machine-learning to identify the diestrous-like stage in both groups of mice. Why use machine learning for one and vaginal cytology for the other?

As “machine learning-defined” diestrus is based on the control vaginal cytology information, the diestrous mice are in fact defined by the same machine learning parameters. We have now noted this.

Specific points(5) With regard to point 2 above, it would be helpful to note the age at which the testosterone samples were taken.

We have included the age in the method.

(6) Lines 198-205 and 258-266: I think these are repeated measures of ANOVA data? If so, report the main relevant effect before the post hoc test result.

We have included the relevant main effect in the manuscript.

(7) Line 415: I don't think the word "although" works in this sentence.

We have changed the wording accordingly.

(8) Lines 514-518: what are the limits of hormone detection in the LCMS assay?

These were originally stated in the figure legend but have now been included in the Methods.

**Reviewer #2 (Public Review):**
SummaryThe authors aimed to investigate the functionality of the GnRH (gonadotropin-releasing hormone) pulse generator in different mouse models to understand its role in reproductive physiology and its implications for conditions like polycystic ovary syndrome (PCOS). They compared the GnRH pulse generator activity in control mice, peripubertal androgen (PPA) treated mice, and prenatal androgen (PNA) exposed mice. The study sought to elucidate how androgen exposure affects the GnRH pulse generator and subsequent LH (luteinizing hormone) secretion, contributing to the pathophysiology of PCOS.Strengths(1) Comprehensive Model Selection: The use of both PPA and PNA mouse models allows for a comparative analysis that can distinguish the effects of different timings of androgen exposure.(2) Detailed Methodology: The methods employed, such as photometry recordings and serial blood sampling, are robust and allow for precise measurement of GnRH pulse generator activity and LH secretion.(3) Clear Results Presentation: The experimental results are well-documented with appropriate statistical analyses, ensuring the findings are reliable and reproducible.(4) Relevance to PCOS: The study addresses a significant gap in understanding the neuroendocrine mechanisms underlying PCOS, making the findings relevant to both basic science and potentially clinical research.Weaknesses(1) Model Limitations: While the PNA mouse model is suggested as the most appropriate for studying PCOS, the authors acknowledge that it does not completely replicate the human condition, particularly the elevated LH response seen in women with PCOS.

We agree.

(2) Complex Data Interpretation: The reduced progesterone feedback and its effects on the GnRH pulse generator in PNA mice add complexity to data interpretation, making it challenging to draw straightforward conclusions.

We agree.

(3) Machine Learning (ML) Selection and Validation: While k-means clustering is a useful tool for pattern recognition, the manuscript lacks detailed justification for choosing this specific algorithm over other potential methods. The robustness of clustering results has not been validated.

Please see below.

(4) Biological Interpretability: Although the machine learning approach identified cyclical patterns, the biological interpretation of these clusters in the context of PCOS is not thoroughly discussed. A deeper exploration of how these clusters correlate with physiological and pathological states could enhance the study's impact.

It is presently difficult to ascribe specific functions of the various pulse generator states to physiological impact. While it is reasonable to suggest that Cluster_0 activity (representing very infrequent SEs) is responsible for the estrous/luteal-phase pause in pulsatility, we remain unclear on the physiological impact of multi-peak SEs on LH secretion, even in normal mice (see Vas et al., Endo 2024). Thus, for the moment, it is most appropriate to simply state that pulse generator activity remains cyclical in PNA mice without any unfounded speculation.

(5) Sample Size: The study uses a relatively small number of animals (n=4-7 per group), which may limit the generalisability of the findings. Larger sample sizes could provide more robust and statistically significant results.

For the key characterization of GnRH pulse generator activity and LH pulsatility in intact PNA mice (Fig.3, 4, 6), we used 6-8 animals in each experiment which we believe to be sufficient. Some of the subsequent experiments do have smaller N numbers and we are particularly aware of the progesterone treatment study that only has N=3 for the PNA group. However, as this was sufficient to show a statistical difference we did not generate more mice.

(6) Scope of Application: The findings, while interesting, are primarily applicable to mouse models. The translation to human physiology requires cautious interpretation and further validation.

We agree.

**Reviewer #2 (Recommendations For The Authors):**
(1) The validation of clustering results through additional metrics or comparison with other algorithms would strengthen the methodology. Specifically, the authors selected k=5 for k-means clustering without providing an explicit rationale or evidence of exploratory data analysis (EDA) to support this choice. They refer to their previous publication (Vas, Wall et al. 2024), which does not provide any EDA regarding the choice of a number of clusters nor their robustness. The arbitrary selection of "k" without justification can undermine confidence in the clustering results since clustering results heavily depend on "k". The authors also choose to use Euclidean distance as the "numerical measure" setting in the RapidMiner Studio's software without justification given the chosen features used for clustering and their properties. The lack of exploratory analysis to determine the optimal number of clusters, "k", to be considered means that the authors might have missed identifying the true structure of the data. Common cluster robustness methods, like the elbow method or silhouette analysis, are crucial for justifying the number of clusters. An inappropriate choice could lead to incorrect conclusions about the synchronisation patterns of ARN kisspeptin neurons and their implications for the study's hypotheses. Including EDA and other validation techniques (e.g., silhouette scores, elbow method) would have strengthened the manuscript by providing empirical support for the chosen algorithm and settings.

It is important to clarify that we did not start this exercise with an unknown or uncharacterised data set and that the objective of the clustering was not to provide any initial pattern to the data. Rather, our aim was to develop an approach that would automatically detect the onset and existence of the key features of pulse generator cyclicity that were apparent by eye e.g. the estrous stage slowing and the presence of multi-peak SEs in metestrous. As such, our optimization was driven by the data as well as observation while retaining the unsupervised nature of k-means clustering. We started by assessed 10 variables describing all possible features of the recordings and through a process of elimination found that just 5 were sufficient to describe the key stages of the cycle. While we appreciate that the use of multiple different algorithms would progressively increase the robustness of the machine learning approach, it is evident that the current k-means approach with k=5 is already very effective at reporting the estrous cyclicity of the pulse generator in normal mice (Vas et al., Endo 2024). Having validated this approach, we have now used it here to compare the cyclical patterns of activity of PNA- and vehicle-treated mice.

(2) The data and methods presented in this study could be valuable for the research community studying reproductive endocrinology and neuroendocrine disorders provided the authors address my comments above regarding the application of ML methods. The insights gained from this work could potentially inform clinical research aiming to develop better diagnostic and therapeutic strategies for PCOS.
**Reviewer #3 (Public Review):**
Summary:Zhou and colleagues elegantly used pre-clinical mouse models to understand the nature of abnormally high GnRH/LH pulse secretion in polycystic ovary syndrome (PCOS), a major endocrine disorder affecting female fertility worldwide. This work brings a fundamental question of how altered gonadotropin secretion takes place upstream within the GnRH pulse generator core, which is defined by arcuate nucleus kisspeptin neurons.Strengths:The authors use state-of-the-art in vivo calcium imaging with fiber photometry and important physiological manipulations and measurements to dissect the possible neuronal mechanisms underlying such neuroendocrine derangements in PCOS. The additional use of unsupervised k-means clustering analysis for the evaluation of calcium synchronous events greatly enhances the quality of their evidence. The authors nicely propose that neuroendocrine dysfunction in PCOS might involve different setpoints through the hypothalamic-pituitary-gonadal (HPG) axis, and beyond kisspeptin neurons, which importantly pushes our field forward toward future investigations.Weaknesses:Although the authors provide important evidence, additional efforts are required to improve the quality of the manuscript and back up their claims. For instance, animal experiments failed to detect high testosterone levels in PNA female mice, a well-established PCOS mouse model. Considering that androgen excess is a hallmark of PCOS, this highly influences the subsequent evaluation of calcium synchronous events in arcuate kisspeptin neurons and the implications for neuroendocrine derangements.

Please see our response to Reviewer 1. It will be important to establish a robust PCOS mouse model in the future that has elevated pulse generator activity in the presence of elevated testosterone concentrations.

Authors also may need to provide LH data from another mouse model used in their work, the peripubertal androgen (PPA) model. Their claims seem to fall short without the pairing evidence of calcium synchronous events in arcuate kisspeptin neurons and LH pulse secretion.

We have demonstrated that ARN-KISS neuron SEs are perfectly correlated with pulsatile LH secretion in intact and gonadectomized male and female mice on many occasions. Given that the pulse generator frequency slows by 50% in PPA mice, it is very hard to imagine how this could result in an elevated LH pulse frequency. While we were undertaking these studies the first paper (to our knowledge) looking at pulsatile LH secretion in the PPA model was published; no change was found.

Another aspect that requires reviewing, is further exploration of their calcium synchronous events data and the increase of animal numbers in some of their experiments.

Please see below.

**Reviewer #3 (Recommendations For The Authors):**
The reviewer believes that this work will greatly contribute to the field and, to provide better manuscript quality, there might be only a few minor and major revisions to be included in the future version.Minor:(1) Line 17: I would change the sentence to "One in ten women in their reproductive age suffer from PCOS" to adapt to more accurate prevalence studies.

We have revised the sentence as recommended.

(2) Line 18 and 19: Although the evidence indeed points to a high LH pulse secretion in PCOS, I would change it to "with increased LH secretion" as most studies show mean values and not LH pulse release data.

While we agree that most human studies show a mean increase in LH, when assessed with sufficient temporal resolution, this results from elevated LH pulse frequency. As such, and to keep the manuscript focussed on the pulse generator, we would like the retain the present wording.

(3) Line 47: Please correct "polycystic ovaries" to polycystic-like ovarian morphology to adapt to the current AEPCOS guidelines.

We have revised the sentence as recommended.

(4) Line 231: Authors stated that "These PNA mice exhibited a cyclical pattern of activity similar to that of control mice" (Figure 5C and D). Please, include the statistical tests here for this claim. Although they say there aren't differences, the colored fields do not reflect this and seem quite different. Could the authors re-evaluate these claims or provide better examples in the figure?

We used Sidak’s multiple comparisons tests for this analysis (as stated in Results). The key data for assessing overall cyclical activity in PNA and control mice is Fig 5B which suggest very little difference. We accept that the individual traces of activity (Fig.5D) do not look identical to controls and, indeed, they are representative of the data set. The key point is they remain cyclical in an acyclic mouse. We have made sure that this is clear in the text.

(5) Subheadings 6 and & of the result section: It sounds confusing to read the foremost claims of the absence of SE differences and next have a clear SE frequency difference in Figures 6 C and D. The reviewer suggests that authors could reorganize the text and figures to make their rationale flow better for future readers.

We have considered this point carefully but find that re-organization creates its own problems with having to use the machine learning algorithm before describing it. It will always be problematic to incorporate this type of data-reanalysis in an original paper but think this present sequence is the best that can be achieved.

(6) Discussion: If PNA female mice did not have elevated testosterone levels, how can the authors compare their results to the current literature? Could this be the case for lacking a more robust ARNKISS neuronal activity output in their experiments? The reviewer recommends a better discussion concerning these aspects.

Please refer to our response to Reviewer #1 comment (2).

(7) Discussion: the authors claim that diestrous PNA mice exhibited highly variable patterns of ARNKISS neuron activity. Would these differences be due to different circulating sex steroid levels or intrinsic properties? Would the inclusion of future in vitro calcium imaging (brain slices) studies contribute to their research question and conclusions? The reviewer recommends a better discussion concerning these aspects.

We have tried to clarify that the highly variable patterns of activity in “diestrous” PNA mice come from the fact that we are actually randomly recording from ARN-KISS neurons at metestrus, diestrus, proestrus and estrus. The pulse generator is cycling but we only have the acyclic “diestrous” smear to go by. This also makes brain slice studies difficult as we would never know the actual cycle stage.

Major:(1) Results section: The reviewer strongly recommends that the LH pulse secretion data for the PPA group be included in the manuscript. If the SEs represent the central mechanism of pulse generation, would the LH pulse frequency match those events? If not, could a mismatch be explained by androgen-mediated negative feedback at the pituitary level? What is the pituitary LH response to exogenous GnRH (i.p. injection) in the PPA group?

Our initial observation showed the frequency of ARNKISS neuron SEs was halved in PPA mice compared to controls. Additionally, one study reported pulsatile LH secretion to be unchanged in this animal model (Coyle, Prescott et al. 2022). Both pieces of evidence clearly indicate that the PPA mouse does not provide an appropriate PCOS model of elevated pulse generator activity. Therefore, we do not see the value of pursuing further experiments in this animal model.

(2) Although the evaluation of relative frequency and normalized amplitude indicate the dynamic over time, the authors should include the average amplitudes and frequencies of events within the recording session. For instance, looking at Figures 1 A and B and Figures 3 A and B, a reader can observe differences in the amplitude due to different scaling axes. Perhaps, using a Python toolbox such as GuPPy or any preferred analysis pipeline might help authors include these parameters.

The amplitude of recorded SEs for each mouse depends primarily on the fiber position. As such, it has only ever been possible to assess SE amplitude changes within the same mouse. It is not possible to assess differences in SE amplitude between mice.

(3) Line 144-156: (Immunoreactivity results): Authors should proceed with caution when describing these results and clearly state that results show a software-based measurement of immunoreactive signal intensity. In addition, the small sample size of the PNA group (N = 4) compared to controls (N = 6-7) seems to mask possible differences. Could the authors increase the N of the PNA group and re-evaluate these results?

We have clarified that the immunoreactive signal intensity is based on software-based measurement. The N number for PNA mice in these studies varies from 4 to 6 depending on brain section availability for the different immunohistochemistry runs. The scatter of data is such that any new data points would need to be at the extreme of the distributions to likely have any impact on statistical significance. As a minor part of the paper, we did not feel that the use of further mice was warranted.

(4) Considering the great variability of PNA's number of SE/hr, the review suggests increasing the N in this group, thus, authors can re-evaluate their findings and draw better analysis/ conclusion.

We have n=6 for the PNA group in the study. As noted above, the variability in SE/hr in Figure 3 comes from assessing the pulse generator at random times within the estrous cycle. Once we separate “diestrous-like” stage for the PNA animals, the variability is decreased as shown in Figure 6.